# Iterative SCRaMbLE for engineering synthetic genome modules and chromosomes

Xinyu Lu[1,2], Klaudia Ciurkot [1,2], Glen-Oliver F. Gowers[1,2], William M. Shaw [1,2,3] & Tom Ellis [1,2] ✉

*Saccharomyces cerevisiae* is closing-in on the first synthetic eukaryotic genome with genome-wide redesigns, including LoxPsym site insertions that enable inducible genomic rearrangements in vivo via Cre recombinase through SCRaMbLE (Synthetic Chromosome Recombination and Modification by LoxPsym-mediated Evolution). Combined with selection, SCRaMbLE quickly generates phenotype-enhanced strains by diversifying gene arrangement and content. Here, we demonstrate how iterative cycles of SCRaMbLE reorganises synthetic genome modules and chromosomes to improve functions. We introduce SCOUT (SCRaMbLE Continuous Output and Universal Tracker), a reporter system that allows sorting of SCRaMbLEd cells into high-diversity pools. Paired with long-read sequencing, SCOUT enables high-throughput mapping of genotype abundance and genotype-phenotype relationships. Iterative SCRaMbLE is applied here to yeast strains with a full synthetic chromosome and histidine biosynthesis modules. Five *HIS* module designs are tested, and SCRaMbLE is used to optimise the poorest performer. Our results highlight iterative SCRaMbLE as a powerful tool for data driven modular genome design.

Over the last 15 years, the field of synthetic genomics has advanced from the initial full synthesis and assembly of genomes to genome minimisation and genome-wide recoding projects in bacteria[1–3]. In eukaryotes, the Synthetic Yeast Genome (Sc2.0) project is nearing completion, with almost all of the 16 synthetic chromosomes completed individually, and many now merged into a single *Saccharomyces cerevisiae* strain[4–6]. But so far, the 'synthetic-ness' of these synthetic genomes has been somewhat limited, with the gene organisation of the assembled chromosomes and genomes always following that of the native version. As we push towards ever-more synthetic genomes, where the genes on chromosomes are organised into functional modules built-to-design from DNA parts[7], we need to develop new methods that allow us to explore the ideal gene layout for a set of genes and to better understand how gene arrangement effects fitness and gene expression.

Recently, we described the development of synthetic genome modules for yeast[8]. Going beyond the work of the Sc2.0 project, these modules are synthetic regions of chromosomes built to contain a set of genes that together encode a common function, with the native copies of those genes removed from the rest of the genome. When the genes in such modules relocate along with their flanking regulatory elements (promoters and terminators), they are known as 'defragmented' modules. But toolkits for modular DNA part assembly in yeast[9–11] allow us to go further and make 'refactored' modules where the native elements like promoters are replaced with well-characterised synthetic counterparts, allowing us to test our understanding of how gene expression control contributes to the function of the module. Indeed, refactoring of endogenous genes is a well-established route to understanding regulation and has been achieved in regions of phage, bacterial and yeast genomes[12–16].

[1]Imperial Centre for Engineering Biology, Imperial College London, London, UK. [2]Department of Bioengineering, Imperial College London, London, UK. [3]Department of Biomedical Engineering, Boston University, Boston, MA, USA. ✉e-mail: t.ellis@imperial.ac.uk

So far, the few studies that have tried defragmenting genes in genomes or refactoring them have not explored how arrangement of the genes effects their expression and function. In yeast, the Synthetic Chromosomal Rearrangement and Modifications by LoxPsym-mediated Evolution (SCRaMbLE) system, developed as part of the Sc2.0 project, offers an exciting tool for this work. SCRaMbLE enables inducible genome rearrangement by Cre-mediated recombination between loxPsym sites, generating gene deletions, inversions and duplications[17–19]. The resulting changes in gene copy number, order and orientation in DNA constructs have been shown to drive transcriptional changes, which in vitro has been demonstrated to fine-tune gene expression to optimise function of a heterologous pathway[20]. However, due to the randomness of the diversity generated by SCRaMbLE, achieving maximal phenotypical improvements in a single round of SCRaMbLE is unlikely and often many rounds are needed. A method known as multiplex SCRaMbLE iterative cycle (MuSIC) was developed to address this, continuously generating genome diversification by SCRaMbLE cycles with screening for the most desired phenotypes done in a stepwise manner[21,22]. We anticipate that iterative rounds of SCRaMbLE (as used in MuSIC) while suitable for finding optimal gene arrangements, will plateau at solutions representing local maxima in the design space. Determining this is challenging, as resolving rearranged genotypes *en masse* is not straightforward with widely used short read DNA sequencing methods.

Most post-SCRaMbLE screening methods are low-throughput and rely on single colony analysis to confirm phenotypes along with genome sequencing to determine the rearranged genotypes and decipher their influence[22–25]. For iterative SCRaMbLE methods that involve extensive screening across multiple cycles, a high throughput method to map all genotypes of a post-SCRaMbLE population and quantify each genotype's relative fitness is lacking. A further complication is that a significant percent of cells in a population of yeast induced to SCRaMbLE do not actually undergo any Cre-mediated rearrangements. These non-recombined yeasts can complicate iterative approaches and use up sequencing and screening capacity. To efficiently isolate SCRaMbLEd cells, a reporter system known as ReSCuES was previously developed to use auxotrophic selection to select against un-SCRaMbLEd cells[25]. ReSCuES flips a dual auxotrophic selectable marker cassette to toggle between two selections, enabling efficient screening of post-SCRaMbLE colonies by implementing targeted selection[25]. However, using this reporter system restricts selectable marker availability in yeast engineering and risks losing positive SCRaMbLE events as the marker is reversible with every Cre rearrangement. To free-up marker usage and maximise the capturing of positive SCRaMbLE events, developing an alternative selection system is necessary.

Here, working in yeast, we explored the idea that iterative SCRaMbLE could be used as a tool for improving the design of refactored synthetic modules and genomes. To test this out, we first constructed synthetic genome modules encoding histidine biosynthesis and examined whether moving these genes into new synthetic arrangements led to growth and function defects. We then showed that SCRaMbLE could be used to rescue a defective *HIS* module by rearranging genes, including inversions, deletions and duplications of genes within this module, with this identifying optimal configurations under specific growth conditions. To facilitate post-SCRaMbLE screening in this work, we developed SCOUT (SCRaMbLE Continuous Output and Universal Tracker) to allow us to efficiently isolate cells using FACS that are expected to have undergone SCRaMbLE. SCOUT was then combined with POLAR-seq[26], to enable us to sequence a sorted pool of cells and resolve the relative fitness of each rearranged genotype by correlating its abundance with phenotype improvements. We then characterised the accumulation of rearrangements during successive rounds of SCRaMbLE, both within a synthetic genome module and across a synthetic chromosome,

demonstrating in both cases how rearrangements improve phenotypes but quickly reach a plateau. Our study gives insights into how gene rearrangements in synthetic yeast can improve selected functions and provides tools for others to use SCRaMbLE to quickly achieve improved phenotypes from combinatorial libraries of rearranged genotypes.

## Results

### Design and construction of the synthetic *HIS* modules

To explore how gene rearrangement can impair or optimise gene expression in synthetic genomes, we first required a testbed system where such changes can be easily linked to a phenotype. In recent work we demonstrated the concept of synthetic genome modules by creating yeast strains with *TRP* modules that co-locate the genes encoding the tryptophan biosynthesis pathway into a synthetic cluster containing loxPsym sites between the genes[8]. Here, we applied our synthetic genome module approach to the genes encoding histidine biosynthesis in yeast, similarly clustering these genes and incorporating loxPsym sites between them. Seven *HIS* genes, namely *HIS1* to *HIS7*, catalyse the 10 reaction steps of the histidine biosynthesis from phosphoribosyl pyrophosphate (PRPP) to L-histidine (Fig. 1A). These genes were relocated to synthetic modules in the *URA3* locus of the yeast genome following one of two approaches; either relocating genes with their native flanking regulatory elements - an approach we called *'defragmentation'*, or by relocating only the protein coding sequences (CDS) of the genes and rebuilding them into a gene cluster with commonly-used modular promoter and terminator parts - an approach we call *'refactoring'* (Fig. 1B).

To relocate the *HIS* genes, we began with a BY4741 strain with the *HIS3* gene partially deleted[27] and used three iterative rounds of CRISPR-mediated deletion to remove the coding sequences of the other six *HIS* genes and their immediate flanking regulatory sequences, being careful to avoid deleting any sequences that might act as regulatory elements of the neighbouring genes (Fig. S1). Each deleted gene was replaced with an individual 23 bp 'landing pad'[14] designed so that it can be easily targeted with CRISPR/Cas9 for future gene restoration or further locus deletion. Successful gene deletion was confirmed by junction PCR from the genomic DNA of transformant colonies (Fig. S2A, B).

To generate a strain with a defragmented *HIS* module, we first constructed the *HIS* genes with their native regulatory sequences as individual gene cassette plasmids (Fig. 1C). These gene cassettes were designed to include around 1 kb upstream and 500 bp downstream sequence for each gene in order to preserve its native promoter and terminator. Synthetic linkers (~200 bp) to connect the 7 gene cassettes into a module were also made by adapting Yeast Toolkit (YTK) assembly connectors[10] and designing these to all embed the 34 bp loxPsym sequence[5,17]. The seven *HIS* gene cassettes, eight synthetic linkers and a *URA3* selectable marker, were linearised from assembled plasmids and co-transformed into the yeast strain yXL010 with all *HIS* genes deleted. During the transformation, all DNA parts were linked together in yeast by homology-dependent recombination, integrating as a 20 kb synthetic module at the *URA3* locus (Fig. S2C). Junction PCR on genomic DNA of the transformants confirmed yeast colonies with the correct defragmented *HIS* module assembled into the desired locus (Fig. S2C–E). One of these clones (yXL052) was selected for further examination.

In parallel to making the defragmented *HIS* module, we also built yeast strains with refactored *HIS* module. To do this, we selected constitutive promoter and terminator parts used in the YTK system[10] and assembled them with *HIS* gene CDS parts to replace their native promoters and 3' UTRs (Fig. 1C). For these modules, we used a variety of promoters, classifying YTK promoters as either high, medium or low strength. We also ensured that we did not repeat any YTK parts within any module designs in order to avoid generating repetitive DNA that

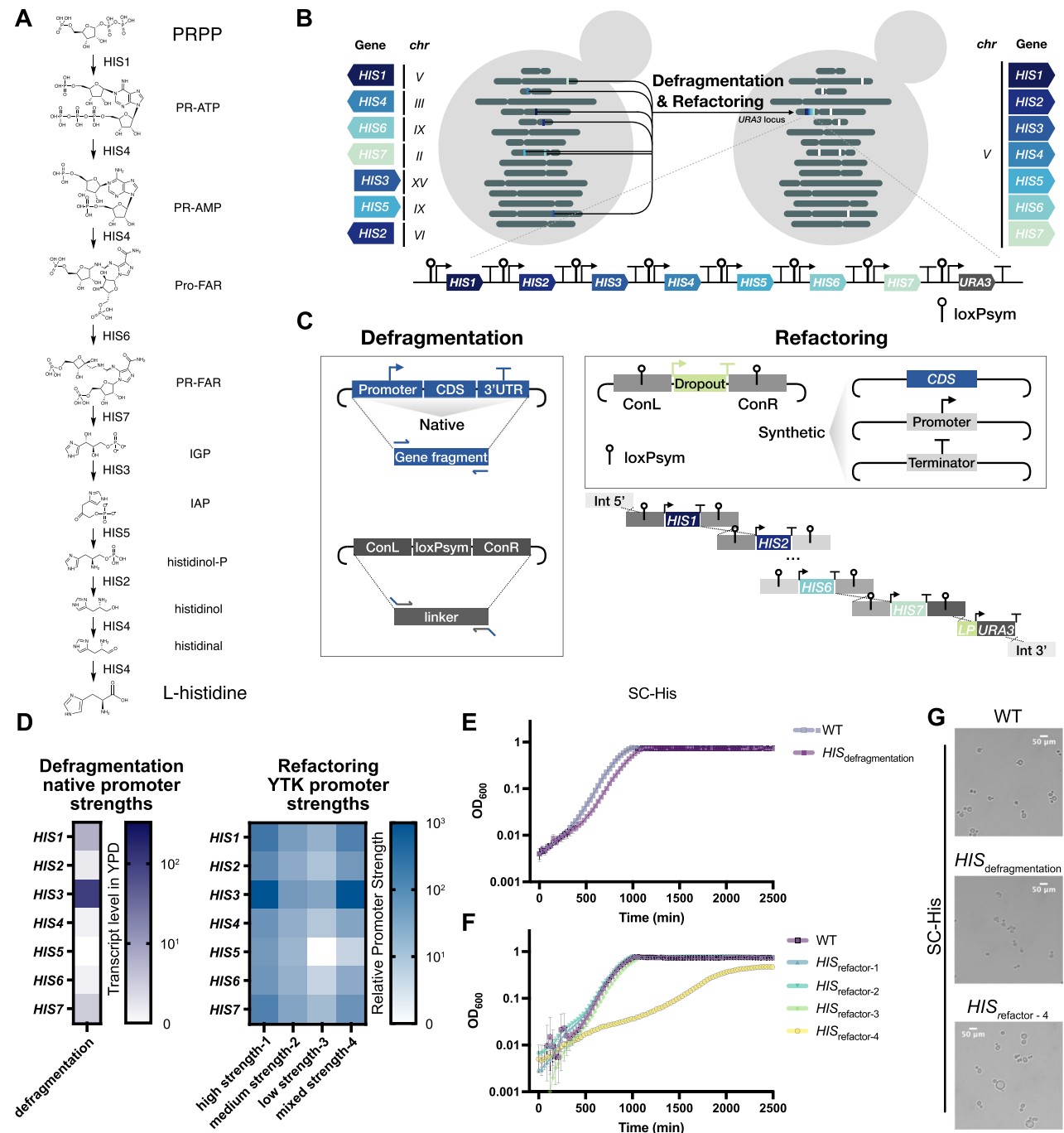

may trigger homologous recombination. We constructed four refactored module designs, with the first three being modules with all strong, all medium and all weak promoters (Fig. 1D). In each case we chose which promoters from each classification should be matched to each of the 7 *HIS* genes by analysing native *HIS* gene transcription levels from previous RNA-seq assays[28] and isoform profiling experiments[29], This meant that the gene associated with the most transcription would be assigned with the strongest promoter from the classification (e.g., the strongest 'weak promoter') and the least transcription would use the weakest promoter.

Finally, to assess the pathway's robustness to a broader range of gene expression variations we made a fourth 'mixed strength' combination where promoters for each *HIS* gene were selected from each of the 3 defined groups, with the strongest promoter assigned for *HIS3*, and one of the weakest promoters assigned for *HIS5* (Fig. 1D), as determined by their respective highest and lowest transcript levels in

RNA-seq data under YPD conditions[29]. As with the defragmented module, these refactored modules were assembled from linear DNA parts in one go by using yeast assembly during transformation to link the DNA together by homology-dependent recombination and insertion into the *URA3* locus of the *HIS* deletion strain (Fig. 1C). A CRISPR-targetable 'landing pad' was included in the module design to enable future module expansion or further editing at this genomic locus. Successful module assemblies were all confirmed by junction PCR from the genomic DNA of transformant colonies (Fig. S3).

## Refactoring with imbalanced *HIS* gene expression causes a functional defect

We next assessed the functionality of the five synthetic *HIS* modules and fitness of the strains by comparing their growth to a wildtype (WT) control strain (yXL014) in rich media (YPD) and synthetic complete media with (SC) or without histidine (SC-His). In these three tested

**Fig. 1 | Design and construction of histidine genome modules. A** Metabolic pathway of L-histidine biosynthesis from 5-phospho-α-D-ribose 1-diphosphate (PRPP) in *S. cerevisiae*. Phosphoribosyl-ATP, phosphoribosyl-AMP, phosphoribosylformiminoAICAR-phosphate, phosphoribulosylformiminoAICAR-phosphate, imidazole glycerol phosphate, imidazole acetol-phosphate are shown as their abbreviations "PR-ATP", "PR-AMP", "Pro-FAR", "PR-FAR", "IGP", "IAP", respectively. **B** Schematic overview of the relocation of *HIS* genes into a synthetic module by defragmentation and refactoring. Seven *HIS* genes, catalysing 10 steps of reactions in the histidine biosynthesis pathway, namely *HIS1* to *HIS7*, including their native regulatory sequences, were partially removed from their native genomic loci and relocated as a module at the *URA3* locus. Genes are labelled in distinct colours. White squares labelled on chromosomes represent deletion of seven *HIS* genes from their native loci and replacement with a 23 bp 'landing pad' which contains a unique CRISPR/Cas9 targeting sequence. **C** Schematic process of the defragmented and refactored *HIS* module assembly. For defragmentation, gene fragments containing the native promoter (1 kb region upstream of the gene), CDS and 3'UTR (500 bp region downstream of the gene) of each *HIS* gene were amplified from the BY4741 genomic DNA and then constructed into entry-level plasmids by Gibson assembly. The linker plasmids were constructed by inserting a loxPsym sequence into the synthetic connectors (ConL and ConR) from the YTK[10]. For refactoring, the CDS of each *HIS* gene was firstly constructed into an entry-level plasmid as a

synthetic 'part' that is compatible for YTK assembly. Next, the YTK promoter, terminator along with the CDS of *HIS* genes were constructed as gene cassettes into the vectors containing linkers embedded with a 34 bp loxPsym sequence. Gene fragments and linkers for defragmentation, and gene cassettes with their linkers for refactoring, were linearised from the plasmids and then integrated into the *URA3* locus as a synthetic module by yeast homologous directed repair (HDR)-based assembly, respectively. **D** Heatmap representing the range of native (purple) and YTK promoter (blue) strengths. Each promoter was assigned for a value from 0-1000 based on the RNA-seq data[29] and YTK[10] characterisation data. Different combinations of promoters were selected for *HIS* gene expression cassettes. **E** Growth curves of the WT control strain (blue) and the strain harbouring the defragmented *HIS* module (purple) in SC-His. Mean $OD_{600}$ from 3 biological replicates are shown as squares, with error bars representing standard deviation. **F** Growth curves of the WT control strain (purple, $n = 4$ biologically independent samples) and the strain harbouring the refactored *HIS* modules (*HIS*$_{refactor-1}$, light blue, *HIS*$_{refactor-2}$, cyan, *HIS*$_{refactor-3}$, light green, *HIS*$_{refactor-4}$, yellow, $n = 3$ biologically independent samples) in SC-His. Mean values are plotted and error bars indicate standard deviation. **G** Microscopy images of the WT control strain and strains harbouring the defragmented and refactored synthetic *HIS* module (*HIS*$_{refactor-4}$) from the overnight culture in SC-His. Source data are provided as a Source Data file.

---

conditions none of the strains except the one harbouring the mixed promoter strength refactored module, known as *HIS*$_{refactor-4}$, exhibited any significant growth defects (Fig. 1E, F, Fig. S4). The *HIS*$_{refactor-4}$ strain showed significantly enlarged budding cells and a slow growing phenotype in SC-His (Fig. 1F, G). This same strain, however, exhibited a normal growth rate and cell size when grown in media with histidine present (Fig. S4B, D). This indicates that the mixed promoter strength module design is specifically sub-optimal for histidine biosynthesis but is unlikely to be affecting cell health in any other way, despite the relocation of the genes.

By matching the promoter choice for each *HIS* gene in *HIS*$_{refactor-4}$ to the corresponding enzymatic reactions of the pathway, we deduced that the slow growing phenotype in SC-His medium might be due to having a very weak promoter (*pRAD27*) for *HIS5* and a very strong promoter (*pTDH3*) for *HIS3*. This creates a possible 150-fold difference in enzyme expression between these two genes that theoretically should lead to the accumulation of the intermediate metabolite imidazole acetol-phosphate (IAP) during pathway function and a large reduction in the flux into the downstream reactions that produce the histidine needed in SC-His conditions. Notably, in *HIS*$_{refactor-3}$, all *HIS* genes were expressed under weak promoters and didn't exhibit noticeable growth defects or morphological changes. The difference in promoter strength between *HIS3* and *HIS5* in *HIS*$_{refactor-3}$ is relatively smaller (~30x) due to the use of the *pRPL18b* promoter for *HIS3* and the *pREV1* promoter for *HIS5*[10]. Given that all our synthetic *HIS* modules contained loxPsym sites between the genes, we hypothesised that this poor design for *HIS*$_{refactor-4}$ could be automatically resolved in vivo by the SCRaMbLE system which can induce gene deletions, inversions, and duplications, through Cre-mediated recombination (Fig. 2A). Thus, this strain (known as yXL216) offers a testbed for using SCRaMbLE as a method to rapidly change gene expression levels and in doing so identify module redesigns that will have improved performance.

**SCOUT enables rapid screening of SCRaMbLEd cells**

Before applying SCRaMbLE to optimise synthetic *HIS* modules, we developed SCOUT (SCRaMbLE Continuous Output and Universal Tracker), a fluorescent reporter system for tracking SCRaMbLE events based on a FLEX Cre-ON switch system[30] (Fig. 2B). In SCOUT, two pairs of heterotypic loxP-variant recombination sites, namely loxP2272 and loxP5171, are placed to flank a version of the *mGFPmut2* gene lacking the ATG start codon. The gene is placed in the antisense orientation with respect to a *TDH3* promoter with ATG start codon. Upon adding

the inducer β-estradiol, Cre-mediated recombination between both pairs of the loxP recombination sites inverts the *mGFPmut2* gene in SCOUT and leading to a DNA construct giving constitutive GFP expression. This recombination also excises one loxP-variant site from each pair, leading to stable and irreversible GFP expression. Cells exhibiting GFP fluorescence, therefore, are indicative of cells with nuclear Cre activity and thus are likely to be associated with SCRaMbLE events occurring in the genomes in those cells. SCOUT is designed as a plasmid-based system to allow us to transiently introduce it into strains ahead of inducing SCRaMbLE and then later remove it from screened strains by culturing in non-selective media. With this capability, SCOUT can be applied in iterative cycles of SCRaMbLE induction and screening.

The SCOUT plasmid was constructed using Golden Gate assembly and made compatible with the YTK[10] and Multiplex Yeast Toolkit (MYT)[11] systems, to allow for easy swapping of DNA parts such as promoters and selectable markers. We initially used a promoter (*pSCW11*) chosen to give SCRaMbLE events exclusively in daughter cells[31]. However, due to the reported leakiness of Cre-mediated recombination when using this strong promoter[19,22,25], we then switched to the weaker *RET2* promoter as this gave reduced leaky expression and the cells still fully converted to GFP⁺ over time (Fig. S5). To confirm whether SCOUT functions as expected, we induced SCRaMbLE in yeast with a defragmented *HIS* module and analysed the phenotypes and genotypes of the post-SCRaMbLE libraries after 4 h of induction. We first confirmed reporter function by screening GFP⁺ cells using flow cytometry and GFP⁺ colonies after plating, showing again the reduced leakiness of the *RET2*-based version, as indicated by fewer GFP⁺ cells and colonies in uninduced samples compared to the *pSCW11* based version (Fig. 2C). Genotyping four randomly selected GFP⁺ colonies revealed SCRaMbLE-mediated gene rearrangements in all cases (Fig. S6).

With SCOUT, we can easily separate SCRaMbLEd cells (GFP⁺) from those without DNA rearrangements using fluorescence-based sorting (FACS). This sorting leaves us with a library of yeast cells with SCRaMbLE events highly likely in synthetic genome modules or other loxPsym-containing regions. When the regions of the genome containing these loxPsym sites are shorter than 35 kb we can use our recently described POLAR-seq method[26] to determine the genotypes of the SCRaMbLEd regions using long-read sequencing and even apply this to the entire post-sorted library.

To determine if such an approach would work for our synthetic *HIS* modules, we took strain yXL052 that harbours the defragmented

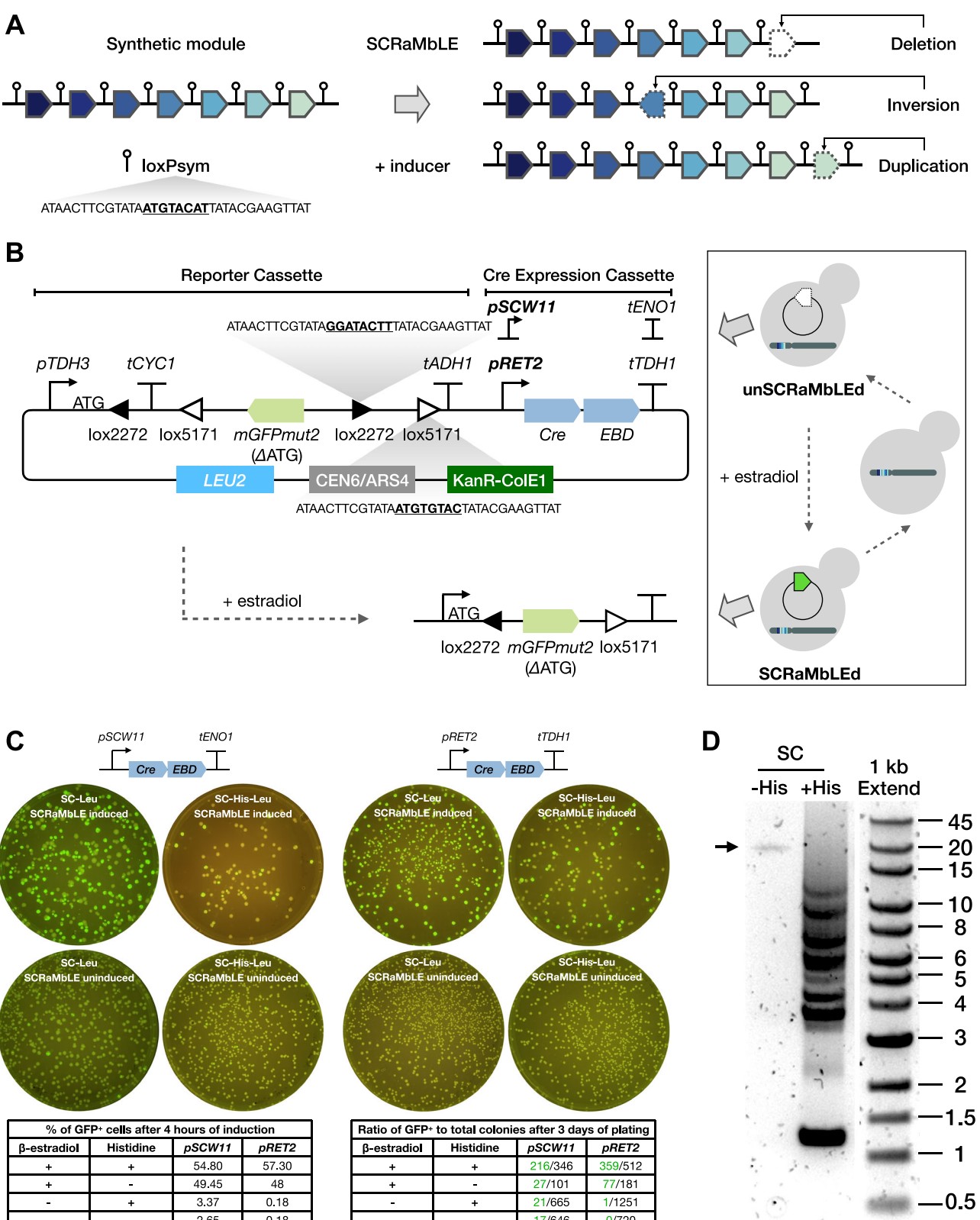

**C**

| % of GFP⁺ cells after 4 hours of induction | | | |
|---|---|---|---|
| β-estradiol | Histidine | *pSCW11* | *pRET2* |
| + | + | 54.80 | 57.30 |
| + | - | 49.45 | 48 |
| - | + | 3.37 | 0.18 |
| - | - | 2.65 | 0.18 |

| Ratio of GFP⁺ to total colonies after 3 days of plating | | | |
|---|---|---|---|
| β-estradiol | Histidine | *pSCW11* | *pRET2* |
| + | + | 216/346 | 359/512 |
| + | - | 27/101 | 77/181 |
| - | + | 21/665 | 1/1251 |
| - | - | 17/646 | 0/720 |

*HIS* cluster, induced SCRaMbLE in a pool of these cells for 4 h in the presence of the SCOUT construct and then used FACS to sort the population for GFP⁺ cells. We then extracted high molecular weight DNA from the post-FACS pool of SCRaMbLEd yeast cells and optimised long range PCR conditions to ensure we could get amplicons from this DNA pool that span the full synthetic *HIS* modules (Fig. S7A–C). When SCRaMbLE and FACS were done in rich media conditions with histidine provided (+His), there was no selective pressure for the yeast to maintain the *HIS* genes. As expected, we saw diverse amplicon sizes arise from the PCR amplification from this yeast library, consistent with the variety of DNA rearrangements that SCRaMbLE provides, including deletion genotypes generated by SCRaMbLE in our library of cells (Fig. 2D). In contrast, when the same experiment was done in media lacking histidine (-His) we observed uniform amplicons around 20 kb

**Fig. 2 | SCOUT for SCRaMbLE screening. A** Schematic of SCRaMbLE inducing rearrangements, such as gene deletion, inversion and duplication between loxPsym sites in synthetic DNA. Genes are labelled in distinct colours. **B** Schematic of the SCOUT reporter design and its application in iterative SCRaMbLE. The SCOUT cassette is an antisense-orientated *mGFPmut2* gene lacking ATG start codon flanked by two pairs of loxP variant sites, loxP2272 and loxP5171. The reporter also encodes a CreEBD expression cassette, where Cre recombinase is fused to the oestrogen binding domain (EBD) so that its function can be induced by β-estradiol. Cre-mediated inversion and excision occur at each pair of loxP variant sites, respectively, resulting in the stable inversion of the *mGFPmut2* gene and deletion of one of each pair of lox sites. The flipped *mGFPmut2* orientation turns on GFP fluorescence. The reporter plasmid can be removed from SCRaMbLEd strains and reintroduced for a new cycle of SCRaMbLE. **C** Assessment of SCRaMbLE reporter (SCOUT) performance. SCRaMbLE was induced by adding 1 µM β-estradiol to a strain containing a defragmented *HIS* module transformed with either of the reporter plasmids. After 4 h of induction, cells were analysed by flow cytometry to determine GFP fluorescence. Percentages of GFP⁺ cells are shown in the left table. Remaining cells were washed twice and plated on SC-Leu and SC-His-Leu plates at a $10^{-4}$ dilution rate. Plates were incubated at 30 °C for 3 days and then imaged under blue light. GFP colony numbers, highlighted in green, were counted and shown in the right table. **D** PCR of a FACS-sorted GFP⁺ cell library to examine the gene rearrangements by SCRaMbLE in a synthetic *HIS* module. Cells were grown in synthetic complete media without histidine (SC-His) and with histidine (SC). PCR primers are designed to target the *URA3* locus, within which the synthetic module is located. Lanes and marker were run on the same gel but were rearranged as indicated. The experiment was independently repeated three times. Source data are provided as a Source Data file.

in length consistent with no deletions of the *HIS* genes in conditions where they are essential (Fig. 2D). These positive PCR results show that post-SCRaMbLE libraries of synthetic *HIS* module containing yeast cells are suitable for POLAR-seq.

## SCRaMbLE and selection enrich for gene duplication in a sub-optimal module

Having established that SCOUT and FACS can generate high quality libraries suitable of POLAR-seq, we next examined SCRaMbLE rearrangements that can improve a poorly designed synthetic genome module. For this we focused on the $HIS_{refactor-4}$ yeast strain that exhibited a major growth defect in the absence of histidine due to the mixed promoter strength refactoring (strain yXL216). The strain was transformed with the SCOUT plasmid (pXL005), generating strain yXL219, and then SCRaMbLE was induced by adding 1 µM β-estradiol. After 4 h of induction, over 1 million GFP⁺ cells were sorted by FACS (Fig. S8) and a subset of these cells was then cultured in SC-His selective media to enrich for phenotypes that can grow well in the absence of histidine (Fig. 3A). We isolated whole genomic DNA from this culture (called 'yXL219$_{pool}$'), amplified the DNA at the *URA3* locus by long range PCR and then sequenced the resulting pool of amplicons following the POLAR-seq method (Fig. S7A).

POLAR-seq identified 127 distinct genotypes from 4,050 annotated full length reads (Fig. S7D). Approximately 86.89% of the total reads displayed genotypes containing 8 transcription units (TUs) within the synthetic *HIS* module, indicating duplication of one gene (Fig. 3B). A small portion of reads (2.57%) had 9 or more TUs. Duplication of *HIS5* was the overwhelmingly dominant trend with it being evident in nearly 90% of all reads (Fig. 3C). The fact that rearrangement and then growth in histidine lacking media selects for cells with duplicated *HIS5* is consistent with our earlier hypothesis that poor $HIS_{refactor-4}$ growth was due to insufficient expression of *HIS5* in this strain.

As well as revealing the abundant genotypes with *HIS5* duplications, POLAR-seq also detected some rare genotypes that would likely be challenging to identify by colony screening methods. Four distinct genotypes, each including two copies of both *HIS4* and *HIS5* TUs, comprised 1.80% of the total reads (Fig. 3D). We also identified two deletion genotypes with some of the *HIS* genes deleted within the module, constituting 0.96% of the total reads. The detection of deletion-containing module genotypes in cells grown in media lacking histidine is surprising. This may just be amplification of modules from dead or non-growing post-SCRaMbLE cells. We did not specifically observe enrichment of shorter amplicons in our data. However, it is important to note that the percentage of each genotype determined by POLAR-seq might not precisely reflect the actual sub-population composition in a population of cells, as there may be enrichment and more sequencing of shorter amplicons, for example, due to PCR bias[32].

To confirm the consistency between genotype abundance from single-cell screening and enriched genotypes identified by POLAR-seq,

as well as their correlation with improved histidine biosynthesis, we picked 14 GFP⁺ colonies at random from the same FACS-sorted library and characterised their growth fitness in SC-His and SC media conditions. We observed a significant increase in the growth rates of all 14 strains in SC-His compared to the parental strain yXL219 (Fig. 3E), demonstrating the fitness improvement post-SCRaMbLE. Strain yXL219$_{SCRaMbLE1-13}$ exhibited the highest maximum growth rate, almost close to that of the WT control. When histidine was added to the growth media, growth rates were similar across all tested strains (Fig. 3E).

Clonal long-read sequencing of the module region for 8 out of the 14 strains revealed a spread of genotypes that aligns well with the POLAR-seq data described above. Notably, the genotypes of 7 of the 8 sequenced strains were among the top three most abundant genotypes identified by POLAR-seq, and all exhibited *HIS5* duplication (Fig. S9A). To determine whether *HIS5* duplication was the key factor rather than point mutations for fitness improvement, we introduced a second copy of *HIS5* gene driven by the same weak *RAD27* promoter at the *LEU2* locus of strain yXL216. The resulting strain showed restored growth fitness in the absence of histidine and similar growth rates across other tested conditions comparing to yXL216 and WT (Figure S10), confirming that the improved growth phenotype was mainly due to the increase of *HIS5* gene dosage, and thus *HIS5* expression. From these results, we confirm that genotype enrichment is indeed likely to be associated with phenotypic improvement in histidine biosynthesis and that the POLAR-seq method accurately captures this.

## A second cycle of SCRaMbLE has limited scope for module improvement

Although a single round of SCRaMbLE may generate significantly improved phenotypes, we have previously seen in other work that function-related phenotypes such as pigment production, can be cumulatively improved through iterative SCRaMbLE[21,22,33]. Thus, to explore the possibility of further improving growth fitness of the strains isolated from yXL219$_{pool}$, we performed a second round of SCRaMbLE on two strains that showed promising results in the first round. These two strains, namely yXL219$_{SCRaMbLE1-2}$ and yXL219$_{SCRaMbLE1-13}$, were chosen because they were the most enriched genotype (Figure S9A) and the fastest growth phenotype (Fig. 3E), respectively. Before starting the second round, the SCRaMbLE reporter plasmid was removed from both strains, and reintroduced fresh through re-transformation (Fig. 3A). The second round of SCRaMbLE was carried out under the same conditions as the first round, generating two post-SCRaMbLE libraries, designated as yXL327$_{pool}$ for yXL219$_{SCRaMbLE1-2}$, and yXL397$_{pool}$ for yXL219$_{SCRaMbLE1-13}$.

In POLAR-seq analysis of the post-SCRaMbLE libraries yXL327$_{pool}$ and yXL397$_{pool}$, we found that the most abundant genotypes in these two libraries kept the genotype of their parental strains yXL219$_{SCRaMbLE1-2}$ and yXL219$_{SCRaMbLE1-13}$, respectively (Fig. S9B and S9C). This suggests that the growth advantage

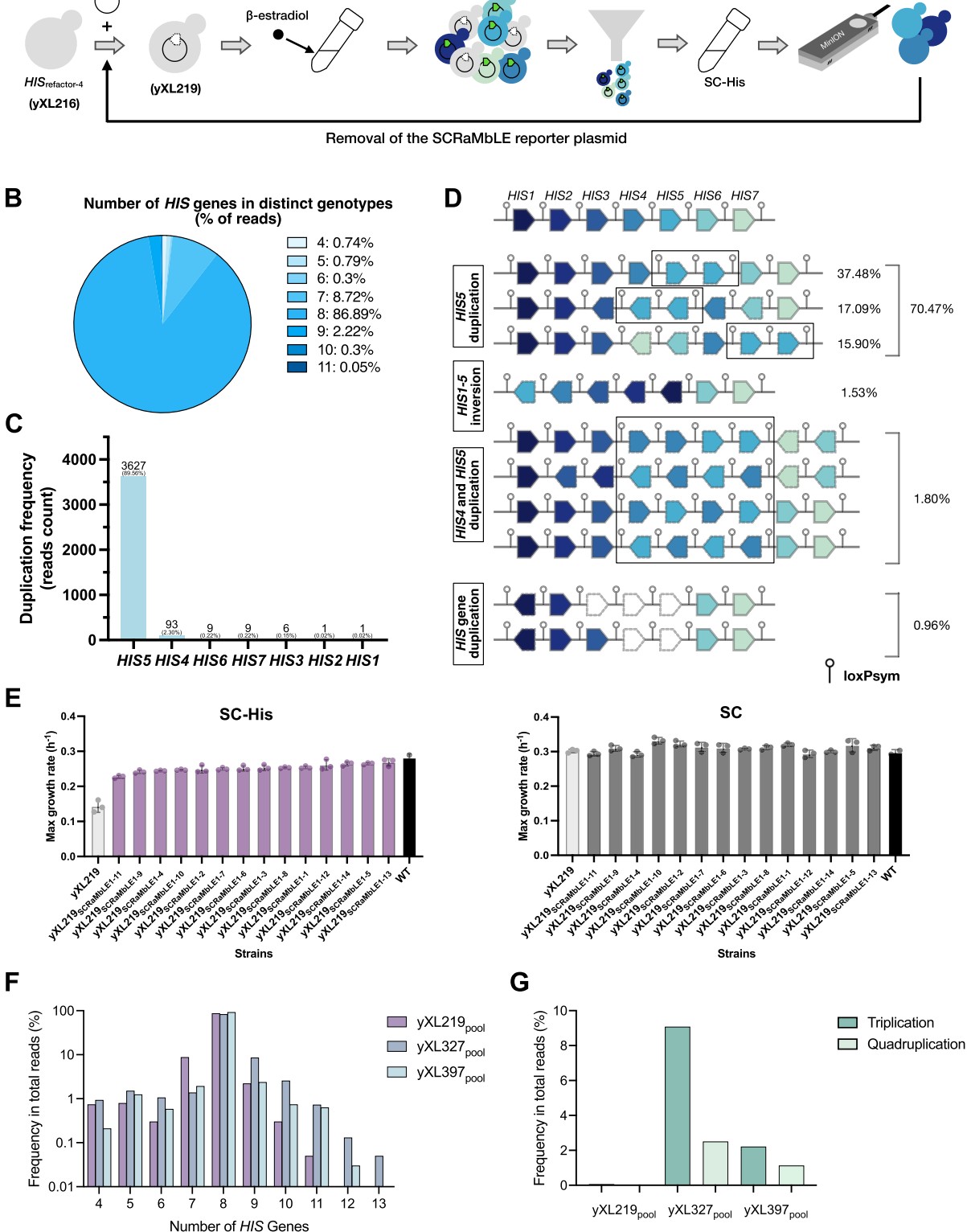

improvements in strain yXL219$_{SCRaMbLE1-2}$ and yXL219$_{SCRaMbLE1-13}$ may have already approached a local optimum in histidine lacking media. In addition, the significantly lower proportion of other genotypes in these cell libraries makes it less efficient to screen colonies showing the non-parental genotype (Fig. S9B, C). Based on the lack of growth improvement, we halted the iterative SCRaMbLE after just two cycles.

We next examined the size variations of the synthetic *HIS* module in our second-round SCRaMbLE libraries. Eight-gene modules with *HIS5* duplication still dominate the population after this round (Fig. 3F and S11). However, we noted a few cases of further expansion in the synthetic *HIS* module, with increased presence of amplicon reads with 11, 12 and 13 genes in the module (Fig. 3F). The largest module we identified had duplications of *HIS2*, *HIS3*, *HIS4* and a quadruplicate of

**Fig. 3 | SCRaMbLE rescues function by duplication of *HIS5*. A** Schematic of SCRaMbLE workflow for synthetic modules. Strain yXL216 that harbours the synthetic *HIS* module (*HIS*$_{refactor-4}$) was transformed with a SCRaMbLE reporter plasmid (pXL005), generating strain yXL219, which can induce Cre expression and indicate SCRaMbLEd cells by GFP fluorescence. Cells showing GFP expression were collected through FACS and subsequently grown in SC-His to enrich improved phenotypes. Genotyping of the post-SCRaMbLE library was performed through POLAR-seq. Top candidates with improved phenotypes were subjected to iterative rounds of SCRaMbLE after curing and re-introducing the SCRaMbLE reporter plasmid (pXL005). **B** Frequency of reads with different numbers of *HIS* genes, identified from the post-SCRaMbLE library yXL219$_{pool}$. **C** Bar chart showing the duplication frequency of each gene in the detected reads from yXL219$_{pool}$. **D** Schematic of representative post-SCRaMbLE genotypes identified in more than 10 reads from yXL219$_{pool}$. Frequency in total reads is labelled on the right in percentage. Rearranged genes are highlighted with dashed lines. Gene duplications are highlighted with black squares. **E** Maximum growth rates of the 14 strains isolated from yXL219$_{pool}$. Bars in grey, purple, and black represents the defective parental strain yXL219, the strains isolated from yXL219$_{pool}$ and the WT control strain (yXL014), respectively. Cultures were grown at 30 °C and performed in 3 biological replicates (WT, $n = 2$ biologically independent samples). Mean values are plotted and error bars represent standard deviation. Culture medium is stated above each graph. **F** Bar chart showing the numbers of *HIS* genes in each distinct genotype identified from yXL219$_{pool}$ (purple), yXL327$_{pool}$ (blue) and yXL397$_{pool}$ (light blue) and their frequency in total reads. **G** Frequency of reads detected with *HIS5* gene triplication (green) or quadruplication (light green) from yXL219$_{pool}$, yXL327$_{pool}$ and yXL397$_{pool}$. Source data are provided as a Source Data file.

the *HIS5*, with a theoretical module size of ~35 kb. We also observed a significant increase in genotypes showing triplication or quadruplication of *HIS5* in the second-round libraries (Fig. 3G). This suggests a continuing enrichment of cells with more copies of the *HIS5* gene when grown under histidine selective conditions. Notably, yXL327$_{pool}$ showed a slightly higher frequency of *HIS5* triplication and quadruplication genotypes than yXL397$_{pool}$ (Fig. 3G), suggesting that differences in parental gene order, positioning and local genetic context may influence Cre recombination frequency and adaptive landscape under selection. Further iterative SCRaMbLE of these genotypes would be interesting to investigate, but unlikely to give significant growth improvements. Overall, both SCRaMbLE and iterative SCRaMbLE on the yXL219 strain revealed that the *HIS*$_{refactor-4}$ module would benefit from being redesigned to have double the expression of *HIS5* (i.e., by use of a stronger promoter for this gene) but that further design changes, such as increased expression of other genes or alternative gene locations and directions would not make any significant difference to function.

## Iterative SCRaMbLE in a synthetic chromosome to maximise phenotype

Having observed SCRaMbLE achieving a solution to poor genome module design in just a single cycle, we wanted to explore whether this was due to the nature of the phenotype we were selecting for or due to the limited possibilities for viable gene rearrangements in the *HIS* module. Presumably, if SCRaMbLE is genome- or chromosome-wide, there are many more viable rearrangements possible and so it may take more cycles to reach phenotypic improvements. To test this, we performed SCRaMbLE across SynV, a whole synthetic chromosome produced for the Sc2.0 project that replaces wildtype chromosome 5 of the *S. cerevisiae* genome. We chose to screen and select for a simple phenotype that doesn't have an obvious solution, as it did in the *HIS* module case. We screened and selected for maximising green fluorescence per cell, when the strain is constitutively expressing sfGFP. To avoid a simple genetic solution, like duplication of the sfGFP gene, we encoded the sfGFP gene on a yeast plasmid (pBAB016) without any loxPsym sites, so it cannot SCRaMbLE. The strain created for this study, synV-pBAB016, is a haploid.

To begin SCRaMbLE, synV-pBAB016 was transformed with the Cre recombinase expressing plasmid *pSCW11-creEBD*. SCRaMbLE was then induced with 1 μM β-estradiol for 4 h, after which time cells were washed twice and plated at various dilutions. A control culture was included that was not induced. After 2 days growth on plates, 3 uninduced control colonies were picked randomly alongside 36 of the brightest post-SCRaMbLE colonies. These were selected by eye under far blue light illumination, with all colonies picked into SC-His-Leu media. Cultures were grown overnight and back diluted 1:100 into fresh SC-His-Leu media in a 96-well plate. Endpoint GFP fluorescence per well was then measured after 48 h of growth. The strain with the highest OD$_{600}$-normalised fluorescence (a proxy for GFP expression

per cell) was isolated. This top performing strain in the first round of SCRaMbLE was termed 'R1'. This strain was next subject to a second round of SCRaMbLE. This process was repeated five times, resulting in strains R0 (pre-SCRaMbLE), R1, R2, R3, R4, and R5 (Fig. 4A).

To directly compare the sfGFP expression of each of these strains, we first removed *pSCW11-creEBD* plasmid from the strains R0-R5 by passaging through two overnight cultures in SC-His media. After confirming the loss of Cre plasmid, we randomly picked 7 colonies for each strain and measured the 48-hour endpoint OD$_{600}$ and GFP fluorescence. We observed a significant increase in sfGFP fluorescence within the first 3 rounds of SCRaMbLE (R0-R3). However, the phenotype did not significantly improve in the fourth and fifth round (R4 and R5) (Fig. 4B). This indicates that either no new SCRaMbLE events could yield further phenotype improvement or that no additional SCRaMbLE events occurred in the final two rounds.

We monitored cell viability by counting colonies for both the uninduced and induced cultures in each round of SCRaMbLE. We calculated the population survival by the percentage of induced colonies compared to uninduced. Population survival was highest for R1 (3.9%) and was lower for all subsequent rounds of SCRaMbLE, suggesting that the presence of existing SCRaMbLE events increased the sensitivity of cell viability to additional SCRaMbLE events (Fig. 4C). This presumably puts an upper limit on the number of SCRaMbLE events that can be accumulated before the cell is non-viable, representing a local maximum in the phenotype-fitness design (Fig. 4D).

To understand how cumulative SCRaMbLE recombination events accumulate phenotype improvements, we sequenced the strains R1-R5 using multiplexed barcoded long-read sequencing. A total of eight samples including R1-R5 were sequenced on R9.4.1 flowcell in a MinION Mk1B device. Run statistics for samples R1-5 are shown in Table S10. By aligning contigs and corrected reads to a pre-SCRaMbLE scaffold sequence, we determined all SCRaMbLE events for each round of SCRaMbLE, as shown in Fig. 4E. As expected, the first round of SCRaMbLE produced the greatest number of SCRaMbLE events, including the inversion of a 114 kb region, which led to the reposition of the centromere and contained two small deletions. R2 and R3 revealed less recombination events than R1. Interestingly, no additional SCRaMbLE events were detected in R4 and R5, suggesting any additional SCRaMbLE events in combination with those present in R3, could not improve the phenotype any further without the loss of cell viability. By comparing the final chromosome rearrangement (R3-R5) to the pre-SCRaMbLEd chromosome (R0), we were able to characterise exactly which genes were affected by each recombination event, as shown in Fig. 4F. This analysis only focused on genes expressing proteins of known function. In summary, over 145,000 bp of the synV chromosome was involved in a recombination event. Within these regions, 5 genes were deleted (*VAB2*, *YEA4*, *FMP10*, *FAU1*, and *PUG1*), two 3' UTR regions were deleted, and eight genes had disrupted 3' UTR sequences (*TIR1*, *RNR1*, *VTC1*, *TIM9*, *YEN1*, *MXR1*, *ERG28*, *MEI4*, and *BMH1*). Aside from potentially *PDA1*, which was inverted and

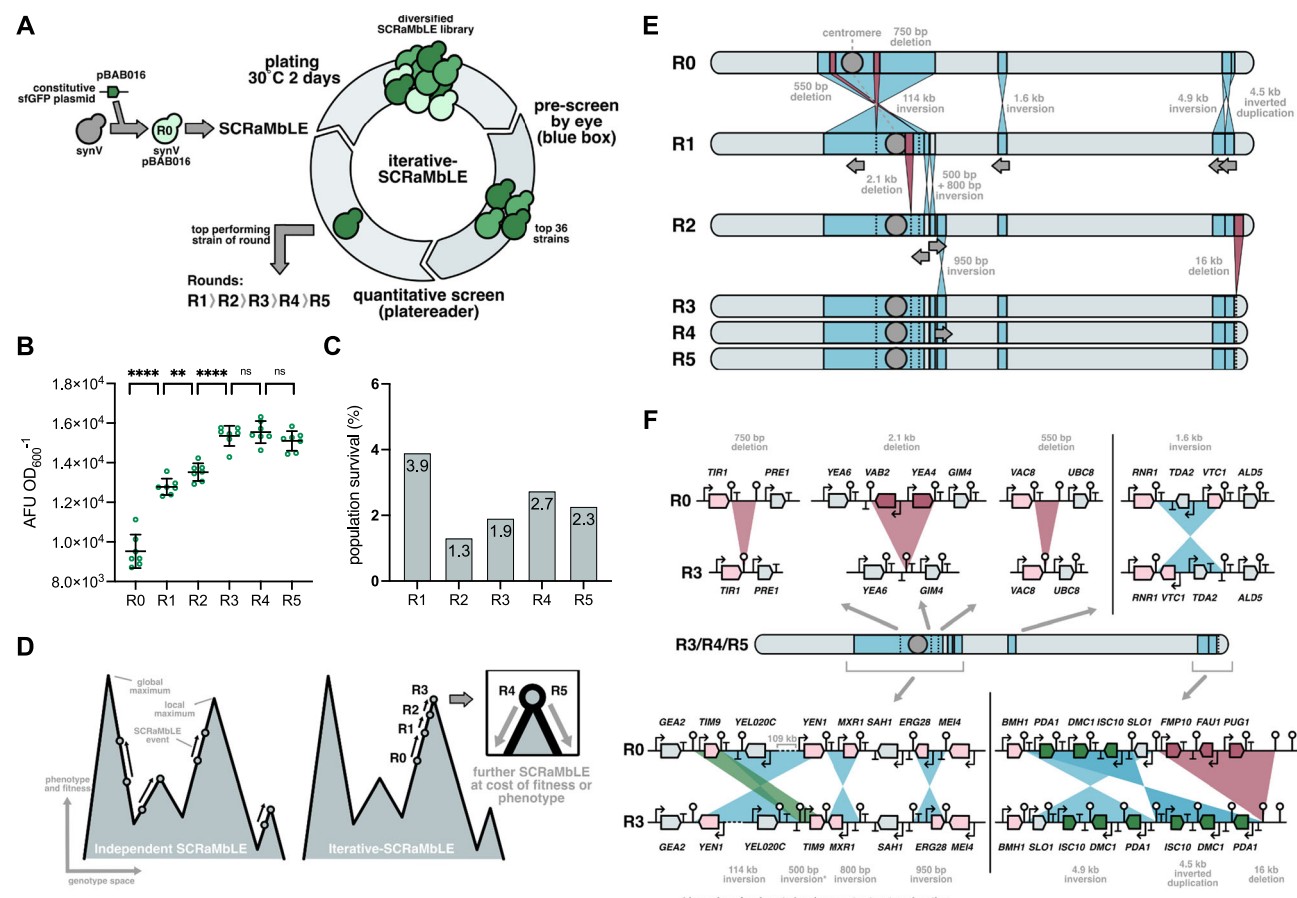

**Fig. 4 | Increasing sfGFP expression by iterative SCRaMbLE of a synthetic chromosome. A** SCRaMbLE was induced in strain synV-pBAB016-*pSCW11-creEBD* (R0) for four hours to create a diversified SCRaMbLE library. The top 36 strains were selected, and fluorescence characterised using a plate reader. The top performing strain (R1) was stocked and subsequently subject to a second round of SCRaMbLE, as before, to produce strain R2, this was repeated for strains R3, R4, and R5. **B** Strains were selected and picked by eye under illumination of blue light ('blue box'). Endpoint 520 nm fluorescence and $OD_{600}$ was measured after 48-hour growth. Mean is shown with error bars that indicate standard deviation for $n = 7$ biologically independent samples. Pairwise comparisons were assessed using one-sided unpaired *t*-tests. R0 vs R1, $p = 9.27 \times 10^{-6}$; R1 vs R2, $p = 0.0073$; R2 vs R3, $p = 1.24 \times 10^{-5}$; R3 vs R4, $p = 0.519$; R4 vs R5, $p = 0.140$. ****$p < 0.0001$; **$p < 0.01$; ns not significant. **C** Colonies were counted for induced and uninduced cultures after each round of SCRaMbLE (R1-R5). Population survival is shown as the percentage of induced colonies to uninduced colonies. For clarity values for each bar are shown as numbers. **D** A simplified two-dimensional depiction of a phenotype and fitness landscape. Transition of a cell (circle) upwards indicates an improvement in fitness and/or phenotype. Arrows indicate individual rounds of SCRaMbLE. During

independent SCRaMbLE experiments many local maxima are sampled but not necessarily reaching the optimum fitness/phenotype within each maximum. Iterative-SCRaMbLE commits a cell lineage to a single local maximum. Once at the optimum fitness/phenotype (depicted here as occurring in R3) no further fitness or phenotype improvements can be seen with subsequent rounds of SCRaMbLE (R4 and R5, inset). Any further SCRaMbLE events, in combination with those already existing result in decreased fitness and/or phenotype. **E** Rearrangement events at each round of iterative SCRaMbLE determined by multiplexed long-read sequencing. Inversions are shown in blue. Grey arrows below each chromosome indicates the new orientation of the region. Deletions are shown in red with the affected region subsequently marked with a dotted vertical line. The centromere is depicted as a grey circle. **F** Summary of SCRaMbLE events in strain R3 (and R4-R5). Deletions are shown in red, inversions are shown in blue, and translocations are shown in green. Pink genes indicate those with a disrupted 3' UTR, red genes indicate those that are deleted, and green genes indicate those that have undergone a copy number change through a duplication event. LoxPsym sites are indicated by white circles on sticks. Source data are provided as a Source Data file.

duplicated when generating R1 and encodes an enzyme that affects carbon supply to the TCA cycle, none of the genes that were affected by any of the SCRaMbLE events observed have an obvious link to sfGFP protein production. Interestingly, none of these recombination events were seen in another previously described strain called VB2 that underwent a single round of SCRaMbLE and showed enhanced GFP fluorescence[34]. This is again consistent with our assumption that the phenotype-fitness landscape is rugged and different improved phenotypes (local maxima) can be achieved through independent SCRaMbLE routes (Fig. 4D).

## Discussion

In this work, we developed a tool, SCOUT, and described methods for iterative SCRaMbLE to be used to improve desired phenotypes in yeast

strains with synthetic genome modules and synthetic chromosomes. To provide a testbed for this work, we also described reorganisation of the 7 genes of the yeast histidine biosynthesis pathway into different designs of synthetic genome modules, including both 'defragmented' and 'refactored' versions. Notably, no obvious fitness defects were observed when defragmenting these genes or when refactoring the genes by using three sets of YTK constitutive promoters with similar strengths, presumably with all these promoters lacking the native regulation of the *HIS* gene promoters. These results confirmed either the robustness of the histidine biosynthesis pathway to alternative levels of gene mRNAs or robustness of the yeast host itself, in terms of being able to tolerate a wide range of HIS enzyme expression levels. Naturally, the yeast histidine biosynthesis pathway is regulated at the protein level by end-product feedback inhibition, and this may explain

why removing native promoters, and thus transcriptional regulation, does not have a major effect on pathway function. Transcriptional regulation of these genes is known to involve the general transcription factor GCN4, transcription factors BAS1 and BAS2(PHO2) and histidine biosynthesis is also co-regulated through transcriptional regulation via intermediate aminoimidazole carboxamide ribonucleotide (AICAR) with purine biosynthesis pathway[35–39]. However, in the limited conditions we assessed cell performance here, we saw no evidence that this native transcriptional regulation was required. Instead, we speculate that in YPD, SC and SC-His media metabolic flux toward histidine biosynthesis and related pathways is likely kept at a functional basal level regardless of *HIS* promoter-encoded features.

While this work and that of others have shown the tolerance of yeast genes to changes in sequence, location and even entire promoters, it is important to note that relocating and refactoring genes may impair cell viability, especially if the genes in question encode an essential function[15,16]. Due to this risk, we propose that synthetic genome module construction should be designed to incorporate some form of in vivo diversification system, such as the SCRaMbLE system used here. This then allows poor performing constructs to quickly be debugged via diversification and growth- or function-linked selection. Similar to our approach, a recent study employed a variant of SCRaMbLE, named Gene Expression Modification by LoxPsym-Cre Recombination (GEMbLeR), but in this case, specifically designed to shuffle only promoters and terminators using orthogonal LoxPsym sites, aiming to diversify gene expression and optimise heterologous pathway function[40,41]. Compared with mutagenesis-based approaches, such as adaptive laboratory evolution (ALE)[42], OrthoRep[43,44] and yEvolvR[45], the SCRaMbLE approach offers an alternative that focuses on adjusting gene expression by changing the copy numbers or orientations of modular DNA regions in the genome. The downstream analysis of the genomic changes leading to functional improvements is therefore more straightforward, as SCRaMbLE only involves region-specific rearrangements, whereas other methods generate more classical mutations like SNPs and Indels that can be harder to identify (especially when genome-wide) and are challenging to interpret phenotypic improvements from. Determining the genotype changes from SCRaMbLE has become relatively straightforward in the last decade with the emergence of long-read DNA sequencing, including targeted methods, like POLAR-seq, that can compare hundreds of SCRaMbLE outcomes at low cost[26]. If duplications of genes within a SCRaMbLEd module are seen as clearly beneficial (as seen in the *HIS*refactor-4 case here), then it informs that a stronger promoter is needed for the genes that are duplicated. Conversely, if genes are lost, then it identifies that they are not needed for the module's function in the assayed conditions. The use of iterative cycles of SCRaMbLE also provides a way to determine whether further module design changes can give significant improvements to module function and cell performance.

Prior work has shown that iterative SCRaMbLE can accumulate phenotype improvements in yeast, such as in heterologous β-carotene or prodeoxyviolacein production[21,22,33]. Here, we took this further and used long-read sequencing over several SCRaMbLE cycles to investigate how cumulative genotype changes led to phenotype improvements from strains with either synthetic modules or entire synthetic chromosomes. In both cases, we observed that the first round of SCRaMbLE yielded strains with notable phenotypical improvements, but subsequent rounds displayed a rapid plateau, indicated by the stable recurrence of the predominant genotype. This is consistent with iterative-SCRaMbLE committing to a single local maximum in the phenotype-fitness landscape. It's important to note, however, that this maximum may not be the global optimum and that it could be limited by the scope of the rearrangements done in the first round. With this in mind, we recommend that the first round of SCRaMbLE is scaled to achieve maximum diversity, for example by rearranging and screening/selecting with a large population. While previous studies typically

performed at least 5 rounds of SCRaMbLE[21,22,33], we found that after the third round, no further viable synV rearrangements were identified that improved GFP production. Similarly, after two rounds of *HIS* module SCRaMbLE and rearrangements screening, parental genotypes predominated within the library, suggesting that additional iterations did not significantly enhance the phenotype and could potentially compromise cell viability. These findings indicate that the optimal SCRaMbLE iterations may be context-dependent, influenced by genomic contexts and the specific traits being optimised. Our work provides insights into the limits and potential of the SCRaMbLE system not only as an optimisation tool but also as a model for adaptive genome evolution.

To quickly and effectively assess the outcome of each round of SCRaMbLE, we developed the SCOUT reporter and combined it with the POLAR-seq method[26], offering ideal tools for applying SCRaMbLE on larger populations. SCOUT combined with FACS enables users to rapidly isolate the population subset likely to have the most SCRaMbLE events as a pool and so aids in enriching the screening/selection step for the most diverse outcomes. This higher sampling depth increases the chance of capturing rare variants such as those with read frequencies below 1%, which might be missed due to throughput limit of traditional screening. The entire screening workflow can be streamlined by saving 3 - 4 days due to avoiding labour-intensive and time-consuming steps such as colony plating, picking and culturing. This works especially well when SCRaMbLE is being used on a synthetic region short enough to generate full length amplicons for POLAR-seq, so that the genotypes in the pool can be sequenced and compared in bulk. When combined with long-read sequencing, SCOUT enables cost-effective, high-throughput genotype analysis by directly revealing synthetic module genotypes from FACS-sorted cell libraries at a cost as low as $1.5 per genotype[26]. The combination of high throughput approaches makes our SCOUT-based pipeline especially valuable for screening of high-diversity populations. In our *HIS* module design, recombination between 8 loxPsym sites can theoretically generate ~1 million distinct genotypes considering only deletions and inversions. While duplication genotypes could further increase this diversity, it is difficult to estimate the upper limit of duplications. In our study, the actual diversity is lower due to the enrichment for faster-growing genotypes under histidine absent selection, in which deletion genotypes were diluted out. We collected >1 million possible SCRaMbLEd GFP⁺ cells via FACS, which should cover a large portion of the diversity space. We recommend optimising screening throughput based on theoretical diversity and biological constraints. Collecting more cells via FACS and reducing following growth time may help cover more diversity and minimise potential bias, depending on experimental purposes.

SCRaMbLE is a rapid, simple, and cheap diversification tool, but applying iterative cycles of SCRaMbLE for metabolic engineering applications and informing synthetic genome design still needs further exploration. We used production of GFP as a selectable phenotype as it is easy to screen. Unlike the case of *HIS* module, it is not directly linked to growth and doesn't have an obvious solution, making it a good benchmark to assess how chromosome-wide modifications across multiple iterative rounds of SCRaMbLE can enhance the heterologous expression of a single protein. This proxy allows us to evaluate applying iterative SCRaMbLE for optimising other functional traits, such as metabolic production and protein secretion. However, parameters for this single-gene model may not be translatable to multi-gene biosynthesis pathways which may involve balancing the expression of multiple proteins and cofactor supply.

To provide insights into how chromosomal rearrangements might drive phenotype improvements, we examined which rearrangements were generated in each round of SCRaMbLE. Our observation of *PDA1* gene duplication from synV SCRaMbLE was also observed in past work, in a synV SCRaMbLEd strain (BC03) that showed improved betulinic

acid synthesis[24]. However, it is not immediately clear to us whether *PDA1* duplication is directly linked to metabolic flux changes that improve heterologous protein production or whether the flanking loxPsym sites are more likely to be involved in SCRaMbLE events. Further investigation is needed to determine whether *PDA1* duplication has a direct biological impact. Another structural rearrangement in synV SCRaMbLE is the 114 kb inversion involving repositioning of the centromere. Similar inversion events containing centromere regions have been seen in SCRaMbLE of synIXR[19] and synXII[25], which changed centromere orientation and positions while maintaining its function. These results are consistent with centromere relocation studies showing that yeast cells exhibit high tolerance to changes in centromere position, as long as sufficient flanking pericentromeric sequences are maintained[46,47]. Centromeres are crucial for chromosome stability and structure. SCRaMbLE offers a platform to generate rearrangements including centromere relocation for identifying viable relocation configurations, facilitating further investigation of the relationship between centromere position and cellular function. Interestingly, comparing with the synV SCRaMbLEd strain VB2 which was generated from a single round of SCRaMbLE and also showed improved sfGFP expression[34], we did not observe any overlap in SCRaMbLE events with R3-5, suggesting that distinct genetic mechanisms may underlie similar phenotypic improvements. This highlights the complexity of genotype-phenotype relationship especially when genotype solutions are non-obvious and hard to predict through rational modelling. Future studies combining our genomic datasets with transcriptomics and metabolomics analysis will help to distinguish the functional and non-functional genotype changes.

A clear link between genotype and phenotype was established by demonstrating that SCRaMbLE generated *HIS5* gene duplication is an effective genetic solution to rescue imbalance of enzyme expression in histidine biosynthesis, highlighting gene duplication as a key mechanism in adaptive evolution. Gene duplication is a fundamental adaptive strategy exists in organisms ranging from microbes to humans[48–52]. When cells are under environmental stress or genetic perturbations, structural rearrangements such as gene duplications can spontaneously occur, increasing the dosage of genes that are important for survival[49,51,53,54]. Given the importance of gene duplications in both natural evolution and laboratory studies, SCRaMbLE serves as a powerful approach for inducibly generating gene duplications through Cre-mediated recombination at loxPsym sites[19,20,22,24,25,55,56]. We showed that tandem duplications of a refactored endogenous gene accumulated extra copies under growth selection pressure during iterative SCRaMbLE, suggesting that SCRaMbLE induced genome structural variation facilitates rapid adaption and improves refactored module function. Beyond simply duplication of genes, other structural variations such as inversions were also observed from SCOUT-POLAR-seq screening. Inversions may influence gene regulation by changing local genomic contexts, altering nucleosome positioning, disrupting cis-regulatory elements, and rearranging the colocalization of genes[57,58]. Future studies should focus on combining our approach with multi-omics approaches to fully understand the mechanistic relationships between these genotype structural variations and the resulting phenotypic changes.

Together, the work described here demonstrates the combinatorial possibilities afforded by synthetic biology when redesigning genomic regions and offers tools and methods to effectively navigate this. Using iterative SCRaMbLE, we have shown that rapid, targeted modular genetic modifications can rescue the functionality of defective synthetic modules and quickly find genotype solutions to phenotype improvements on a chromosome-wide scale. Our study also demonstrates how genome refactoring can be done at the module scale and still achieve efficient function for a key metabolic pathway, and it shows how the approach can efficiently obtain genotype-to-phenotype datasets from POLAR-seq that are valuable for determining

ideal gene expression profiles for cells grown in different conditions. By comparing rearrangement patterns across different conditions and through iterative cycles, we will be able to model how these synthetic modules behave under various conditions. This will inform the development of dynamic control systems that can fine tune gene expression to optimise pathway performance in response to environmental changes. Future studies focusing on systematically exploring more complex structural rearrangements, including inversions of modular DNA regions and centromere relocations will help us understand more about their influence on gene expression and chromosomal stability. Combining transcriptome analysis for these rearrangement patterns will provide insights in how transcriptional behaviours interact with DNA sequence contexts. These datasets will help predict the optimal gene layout for designing synthetic modules and genomes tailored to specific functions. We anticipate that datasets like these will be crucial for the future of designing custom modular synthetic genomes.

## Methods

### Strains and media

Yeast strains generated in this study are derived from BY4741 yeast (*MATa his3Δ1 leu2Δ0 met15Δ0 ura3Δ0*)[27], including the Sc2.0 project strain synV-pBAB016[34]. See Supplementary Table 1 for a full list of derived yeast strains. NEB Turbo competent *Escherichia coli* (*E. coli*) from New England Biolabs (NEB) was used for all DNA cloning and plasmid propagation work.

Yeast extract Peptone Dextrose (YPD) media (10 g L$^{-1}$ yeast extract (VWR), 20 g L$^{-1}$ peptone (VWR), 20 g L$^{-1}$ glucose (VWR)) was used for general culturing of yeast cells, unless otherwise stated. Synthetic Complete media (SC; 6.7 g L$^{-1}$ Yeast Nitrogen Base without amino acids, 1.4 g L$^{-1}$ Yeast Synthetic Drop-out Medium Supplements without L-uracil, L-tryptophan, L-histidine, L-leucine, 20 g L$^{-1}$ glucose) was used for auxotrophic selection experiments, or was used with all amino acids supplemented as a defined complete medium. Amino acids such as 20 mg L$^{-1}$ L-tryptophan, 20 mg L$^{-1}$ L-histidine, 20 mg L$^{-1}$ uracil and 120 mg L$^{-1}$ L-leucine were supplemented into SC media depending on the required auxotrophic selection. For growth on plates, media were supplemented with 20 g L$^{-1}$ bacto-agar (VWR). Unless otherwise stated, all other components used in the media were supplied by Sigma Aldrich.

Luria-Bertani (LB) medium was used for culturing *E. coli*. LB agar was prepared by dissolving 37 g L$^{-1}$ LB agar powder (VWR) into required amount of distilled water. Antibiotics such as ampicillin (100 μg mL$^{-1}$), chloramphenicol (34 μg mL$^{-1}$), kanamycin (50 μg mL$^{-1}$) and spectinomycin (100 μg mL$^{-1}$) were supplemented when necessary.

### Plasmids

A full list of plasmids generated in this study can be found in Supplementary Table 2 (gap repair donor and gene fragment plasmids), Supplementary Table 3 (linker plasmids), and Supplementary Table 7 (pre-assembled linker vectors). Unless otherwise stated, all plasmids were constructed using the MoClo Yeast Toolkit (YTK)[10]. Standard parts from the YTK libraries, such as promoters, terminators, and plasmid backbones, were typically used, with these stored in entry vectors by the group. For parts not found in the group's standard libraries, DNA was typically amplified by PCR using primers that add the necessary YTK-compatible overhangs so that the PCR product can be directly used in its intended YTK assembly. A few parts were kindly provided by other Ellis Lab members, see acknowledgements in the parts list (Supplementary Table 6). These parts were then assembled into cassettes (listed in Supplementary Table 8) and the cassettes were then assembled into multigene cassettes (listed in Supplementary Table 9) through the standard YTK assembly workflow[10]. For a full list of oligos and their sequence, see Supplementary Table 4.

Golden Gate Assembly was used to assemble plasmids whenever the DNA sequences being assembled are free of recognition sites for

the type IIs restriction enzymes used by the Golden Gate reaction, i.e., BsmBI, BbsI, or BsaI. Gibson assembly was used when the sequence of DNA fragment to be assembled was not free of recognition sites of the type IIs restriction enzymes BsmBI, BbsI, and BsaI.

## Genomic DNA and plasmid extraction for PCR

Genomic DNA of yeast was extracted following a LiOAc–SDS isolation protocol[59]. Briefly, 200 μL of overnight yeast culture was lysed in a LiOAc–SDS solution (200 mM LiOAc, 1% (w/v) SDS). DNA was then precipitated with ethanol and the resulting pellet was resuspended in 100 μL nuclease-free water. Plasmids were extracted using the QIAprep Spin Miniprep Kit (Qiagen) according to the manufacturer's instructions. PCR amplification was performed using Q5 High-Fidelity DNA polymerase (NEB) in a 50 μL reaction, following the cycling conditions recommended by the manufacturer, with the extracted genomic DNA or plasmid as the template.

## Yeast transformation

A colony was picked out from the plate and grown to saturation in 2 mL appropriate media overnight (30 °C, 250 rpm). The next day, cell culture was diluted to $OD_{600}$ - 0.2 in a 50 mL Falcon centrifuge tube with 10 mL fresh media and grown for ~6 h to $OD_{600} = 0.8$–1.0. Cells were pelleted by centrifuging at room temperature ($2000 \times g$, 10 min), then washed once with 10 mL 100 mM lithium acetate (LiOAc, Sigma Aldrich). Centrifugation was repeated and cell pellet was resuspended in ~600 μL 100 mM LiOAc. 100 μL of this mixture was added to 64 μL DNA cocktail containing 10 μL of boiled salmon sperm DNA (ThermoFisher) per transformation, and then gently mixed with 296 μL PEG-3350/LiOAc mixture (260 μL 50% (w/v) PEG-3350 and 36 μL 1 M LiOAc). This mixture was then placed in a heat block at 42 °C for 40 min and cells were pelleted by centrifugation at $6200 \times g$ for 1 min. Pellets were resuspended in 100–200 μL 5 mM $CaCl_2$. After a 10 min recovery, cells were plated onto the appropriate agar media for selection.

## CRISPR/Cas9 genome engineering

For multiplex gene deletion, 50 ng of the CRISPR/Cas plasmid (pWS2081-*URA3*), 600 ng of each sgRNA plasmid were mixed together with 0.5 μL BpiI (ThermoFisher), 1 μL of 10X Buffer G (ThermoFisher), and nuclease-free water to make up to 10 μL. This mixture was then incubated at 37 °C for 8 h followed by 80 °C heat inactivation for 10 min. 5 μg of each donor DNA was added to this mixture to a total volume of 64 μL to be used for the yeast transformation. Donor DNA was generated by PCR amplification from the genomic DNA and purified by DNA Clean & Concentrator Kit (Zymo Research). gRNA plasmids were constructed by T4 PNK phosphorylating (NEB) and annealing two oligos, followed by a BsmBI Golden Gate assembly to insert the small fragment into the SpCas9 sgRNA Dropout vector (Ellis lab plasmid pWS2069). Oligos for gRNAs were designed by adding the sequence 5′-AGAT-3′ at the 5′ end of the guide sequence and 5′-AAAC-3′ at the 5′ end of the reverse complementary of the guide sequence.

Unless otherwise stated, for genome integration and other genome editing experiments, 250 ng of the CRISPR/Cas plasmid and 500 ng of each DNA fragment was combined with 10 μL boiled salmon sperm DNA, made up to 64 μL with nuclease-free water to be used for the yeast transformation. DNA fragment was generated by PCR amplification and purified by DNA Clean & Concentrator Kit (Zymo Research). gRNA plasmids were constructed according to the 'gRNA-tRNA Array Assembly' methods described in the Multiplex MoClo Toolkit[11]. Briefly, each gRNA-tRNA fragment was PCR amplified, gel purified and assembled into corresponding gRNA-tRNA vector using Golden Gate assembly.

For a full list of gRNAs used for CRISPR/Cas9 genome engineering, see Supplementary Table 5.

## Plasmid curing

For curing *URA3* containing plasmids, 5-FOA (5-fluoroorotic acid) counterselection was used[60]. Colonies were inoculated into 2 mL YPD media and grown overnight (30 °C, 250 rpm). This culture was streaked using a 10 μL loop onto the agar plate supplemented with 5-FOA (Formedium). Colonies showing growth after incubation for 3 days at 30 °C suggested successful plasmid curing.

For an auxotrophic marker that is not *URA3*, strains were firstly streaked onto the YPD agar plate. After 3 days of incubation at 30 °C, colonies were picked and inoculated into 2 mL of fresh YPD media and grown overnight (30 °C, 250 rpm). This culture was then streaked using a 10 μL loop onto the YPD agar plate and incubated at 30 °C for 3 days. Colonies were then streaked onto both the agar media plates with selection and also agar media plates lacking selection. Strains showing growth on the media lacking selection but not the media with selection suggested successful plasmid curing.

## SCRaMbLE

Strains transformed with either pXL004, pXL005 (SCRaMbLE reporter plasmids) or plasmid *pSCW11-creEBD* were grown overnight in 2 mL of appropriate media (30 °C, 250 rpm). This culture was diluted in 5 mL of fresh media to $OD_{600}$ - 0.2 and grown for ~4 h to $OD_{600} = 0.4$–0.5. SCRaMbLE was induced by adding 1 μL of 5 mM β-estradiol dissolved in DMSO to the culture. For the control (uninduced), 1 μL of DMSO was added. Cultures were grown for an additional 4 h before being washed twice in water and resuspended in 1 mL PBS. Cells were diluted $\times 10^{-3}$ and $\times 10^{-4}$ and plated onto appropriate agar plates and grown for 2–3 days at 30 °C.

## Iterative-SCRaMbLE

For the iterative SCRaMbLE of synthetic *HIS* modules, after each round of SCRaMbLE and POLAR-seq, the phenotypes and genotypes of several of the most abundant strains were characterised individually. The top performers were then grown in rich media to remove the SCRaMbLE reporter plasmid (pXL005), in which the sfGFP gene is reversed. The strains confirmed with the removal of this plasmid were then reintroduced with a new pXL005 and subjected to the next round of SCRaMbLE.

After each SCRaMbLE round of the strain synV-pBAB016, 36 SCRaMbLE colonies and 3 control colonies were picked into 500 μL selective media in a 2.2 ml deep well 96-well plate (Grenier). After a single overnight growth (30 °C, 250 rpm) 1 μL was used to inoculate 99 μL of appropriate selective media and $OD_{600}$ and endpoint fluorescence measured in a plate reader at 520 nm after 48 h of shaking growth at 30 °C. The strain with the highest $OD_{600}$-normalised fluorescence was streaked out and used to inoculate 5 mL of fresh media for a subsequent round of SCRaMbLE as well as being stocked in 25% glycerol at −80 °C. Once a high performing strain was obtained for each round of SCRaMbLE these top strains were streaked onto appropriate selective media lacking selection for curing the Cre-recombinase plasmid.

## Flow cytometry

The fluorescence of cells was measured by an Attune NxT Flow Cytometer (ThermoFisher). The following settings were used for measuring the size of the cell, complexity of the cell and fluorescence of the cell: FSC 100 V, SSC 355 V, BL1 450 V. 10,000 events of yeast population gated by forward and side scatter were collected for each experiment and analysed by FlowJo.

## Fluorescence-activated cell sorting (FACS)

FACS sorting was performed on the BD FACSAria III Cell Sorter (BD Biosciences) to select yeast cells with GFP fluorescence. Yeast cells, washed and resuspended in PBS buffer post-SCRaMbLE, were

transferred to a 5 mL FACS tube (Invitrogen) and diluted to appropriate density with PBS buffer for FACS sorting. The 70 μm nozzle was selected for the sorting. GFP$^+$ cells sorted from the FACS instrument were collected in a 15 mL Falcon centrifuge tube. A subset of cells was immediately inoculated into SC-His and grown for 2–3 days to reach saturation (30 °C, 250 rpm). The remaining cells were spun down at 3739 × g for 20 min. The pellet was resuspended in 0.25 mL PBS and stocked in 25% glycerol at −80 °C.

For the FACS analysis, yeast cells were firstly selected based on morphology (FSC-A vs SSC-A) to select for intact, healthy yeast cells with characteristic yeast cell size while excluding debris. The main population as a dense cluster was gated first. Single cells were then selected based on a double doublet-discrimination (FSC-A vs FSC-H and SSC-W vs SSC-H). Single GFP$^+$ cells were then selected based on GFP expression (FSC-A vs GFP). GFP expression was selected with the 530/30 bandpass filter.

### High molecular weight (HMW) genomic DNA isolation

HMW genomic DNA of the post-SCRaMbLE cell libraries was isolated according to the method published by Denis et al. [61] with the following modifications in the protocol: yeast culture of the post-SCRaMbLE library was grown in 50 mL of appropriate media overnight until $OD_{600}$ = 5–15, Zymolyase (Zymo Research) was replaced with Lyticase (Sigma Aldrich, 600 U per 1 mL of $OD_{600}$ = 1), and all centrifugation steps were performed at 4000 × g. HMW genomic DNA of the individual strains was isolated from 2 mL of overnight yeast culture using the adapted protocol and proportionally reduced reagents.

HMW genomic DNA of the post-SCRaMbLE synV-pBAB016 strains R1-R5 was isolated for nanopore sequencing using Genomic-tip Kits (Qiagen) according to the instructions from the manufacturers. Any handling of DNA was done using wide bore tips where appropriate.

### POLAR-seq

POLAR-seq and data analysis were performed according to the method published by Ciurkot et al. [26]. Briefly, the reads covering the full synthetic *HIS* module are selecting based on presence of primer binding regions using Porechop [62]. Reads shorter than the size of the smallest amplicon detected on agarose gel are removed. Genotypes are revealed by annotating each read with Liftoff [63]. Genotypes were visualised using DnaFeaturesViewer [64].

Long range PCR of the full synthetic *HIS* module was set up in a 25 μL of LA Taq Hot Start (TaKaRa, Shiga Japan) PCR reaction according to the manufacturers' protocol, using 20 ng of genomic DNA as the template and primers at a final concentration of 0.1 μM. Amplicon was verified by gel electrophoresis (50 V for 5 h) on a 0.6% agarose gel.

The pool of amplicons was prepped for sequencing with NEBNext Companion Module and the ligation kit SQK-LSK109 or SQK-LSK114. The library was sequenced on Flongle (FLG001 or FLG114) using MinION Mk1B. The data was collected with the latest version of Min-Know (22.05.5–23.04.6).

### Nanopore sequencing of the post-SCRaMbLE synV-pBAB016 strains (R1-5)

Genomic DNA was quantified using a Qubit 2.0 fluorimeter to ensure a minimum amount of 1 μg present in at least 50 μL volume (20 ng/μL). The DNA library was prepared using the SQK-LSK109 kit with 1D native barcoding EXP-NBD104 kit (Oxford Nanopore Technologies). After ligation of the barcoding sequences BC01-05 to samples R1-R5, respectively, samples were pooled, along with 3 other barcoded samples. A total of eight samples including R1-R5 were sequenced on a single R9.4.1 flowcell in a MinION Mk1B device. Sequencing was allowed to proceed for 48 h using the latest version of MinKnow software with local basecalling.

Raw reads were corrected using the correction step of canu v1.8 (www.canu.readthedocs. io) to create a set of long reads (N50 ~ 45 kb) with a higher read accuracy and sequencing genome coverage of 40x. De novo contigs were generated using these corrected reads with smartdenovo (https://github.com/ruanjue/smartdenovo) and both contigs and corrected reads were aligned to the pre-SCRaMbLE synV sequence using LASTAL. Alignments were viewed in Tablet with a synV GFF3 file to identify recombination events with boundaries at loxPsym sites [65]. To better facilitate multiplexing in sequencing analysis each bioinformatic programme was run through a bash shell script to automatically process R1-R5 reads sequentially.

### Plate reader assay

Overnight cultures were harvested, washed and used to inoculate 100 μL cultures in a 96-well plate with a starting $OD_{600}$ normalised to 0.02. Plates were incubated and measured in a Synergy HT Microplate Reader (Biotek) shaking at 30 °C. Mean absorbance values of equivalent blank media wells were subtracted from data points.

### Microscopy

A single yeast colony was used to inoculate 2 mL of appropriate media. Cultures were grown overnight (30 °C, 250 rpm) and visualised on a Nikon Eclipse Ti inverted microscope at 20x magnification and optical conf. Bright field (BF) images were captured using the Nikon NIS-Elements Microscope Imaging Software. Fiji [66] was used to process the images and add the scale bars.

### Statistics and reproducibility

Experiments that created quantitative data were performed in biological triplicates (*n* = 3) or greater, except for the WT strain in Fig. 3E (*n* = 2 biological independent samples) due to well constraints in the single-plate assay. Consistent growth of this WT strain was confirmed in other experiments (*n* = 3 biological independent samples). Yeast colonies were chosen at random from plates. Growth curves, pie charts and bar charts in the figures are generated using GraphPad Prism (v9.5.0). Pairwise comparisons were assessed using one-sided unpaired t-tests. Mean values were plotted and error bars represent standard deviation. ****$p < 0.0001$; **$p < 0.01$; ns, not significant. Long PCR experiment (Fig. 2D) was independently repeated three times. No data were excluded from the analyses.

### Reporting summary

Further information on research design is available in the Nature Portfolio Reporting Summary linked to this article.

## Data availability

The source data underlying Figs. 1D–F, 2D, 3B–G, 4B, C and all Supplementary Figs. are provided as a Source Data file. All raw reads generated using Oxford Nanopore sequencing are deposited in the Sequence Read Archive (SRA) database under the BioProject accession number PRJNA1284883. All other data are presented in the main text or the supplementary materials. Source data are provided with this paper.

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

## Acknowledgements

This research was supported by a Chinese Scholarship Council (CSC) PhD scholarship to X.L. and Wellcome Trust Discretionary Award (221267/Z/20/Z) providing funding for K.C. and T.E. W.M.S. and G-O.F.G. were supported by BBSRC CASE PhD studentships.

## Author contributions

X.L., K.C., G.-O.F.G., W.M.S. and T.E. designed the experiments. X.L., K.C., G-O.F.G. and W.M.S. performed the experiments. X.L., K.C., G.- O.F.G. and W.M.S. performed the data analysis. X.L., K.C., G.-O.F.G., W.M.S. and T.E. interpreted the results. X.L. and T.E. prepared the manuscript. All authors reviewed and approved the final manuscript.

## Competing interests

K.C. is now an employee of Oxford Nanopore Technologies but was solely employed by Imperial College London during the time generating the data included in this paper. All other authors declare no conflicts of interest.
