## [Transparent Peer Review file · Nature Communications]

Iterative SCRaMbLE for Engineering Synthetic Genome Modules and Chromosomes

Corresponding Author: Professor Tom Ellis

Version 0:

Reviewer comments:

Reviewer #1

(Remarks to the Author)

The manuscript "Iterative SCRaMbLE for Engineering Synthetic Genome Modules and Chromosomes" by Lu et al. describes iterative SCRaMbLE approaches on a genome module and chromosome level. A novel concept for targeted selection and analysis of scrambled genome modules named SCOUT was introduced. The method is well substantiated with respective data sets and was applied on a histidine biosynthesis genome module as a proof of concept. The methodology has novelty and will provide the community with a valuable new tool to apply SCRaMbLE based methods in more straight forward iterative manner, paving the way for pathway and genome diversification and screening. Some additional considerations and motivation for conducted experiments would benefit the coherence of the study. For example, it would have been nice to see the application of the SCOUT system on the chromosome level. Did the authors consider this and if not why? Also, novelty aspects of the study should be highlighted and benchmarked a bit more compared existing studies.

Further considerations and remarks are summarized as follows:

Line 101: could be a more explicit what "rearranging genes within this module" would entail (e.g. also including duplications).

Line 123-127: For clarity one could mention here already that loxP sites are also included.

Line 142: General comment, good to introduce abbreviation when used first time, e.g. here YTK. Could also consider having a list of relevant abbreviations (e.g. for MYT, IAP, etc.)

Line 191/192: good to make a bit clearer the motivation behind assigning the strongest promoter for HIS3 and the weakest promoter assigned for HIS5

Line 205-210: Would be good to elaborate a bit more why HISrefactor-4 behaves so much differently from low strength-3 when it comes to growth. HIS3 and HIS5 difference in expression level seems to be similar.

Line 233: "excises one loxP-variant" instead of "excises of one loxP-variant"

Line 241 onwards: good to introduce the estradiol inducibility of the system to ease overall understanding

Line 276-278: consider rephrasing "constructed for Figure 1", and be a bit more explicit, e.g. which cell with which type of module was chosen?

Line 303: Were any point mutations observed in the sequence data?

Line 315: Could the deletion be connected to that the entire module was relocated? A simple PCR targeting parts of the module could give insights if it is still present.

Line 319: Is enrichment of shorter amplicons what you observed here?

Line 321: The approach of picking random colonies would not give direct insights into all relevant genotypes observed using POLAR-seq. Why not choosing a strategy to analyze the most represented genotypes and evaluate these instead of randomly picking some (e.g. picking representative examples of each type shown in Fig.3 D)?

In addition, cleaner experiments would have been to reintroduce the modified His module into the initial strain. This would exclude potential secondary effects e.g. in connection to random mutations in the genome. Was this considered?

Line 329 and Fig. S9A: Only 8/14 of the clones were further investigated in terms genotype. Further insights on 1-4, 1-5, 1-6, 1-7, 1-9 and 1-14 would be interesting to discuss as well. Especially for the ones which were scoring with relatively high growth rates in Fig. 3E SC-His (e.g. 1-14).

Line 420: Generally, why was not FACS considered to enrich for high FL variants? By randomly picking colonies, not the full scope of variants can be screened/analyzed. Could higher FL variants have been identified?

Line 487: Consider rephrasing “synthetic promoters” as the promoters used here are not really “synthetic”

Line 496: explain abbreviation AICAR

Line 509: introduce abbreviation GEMbL2R

Line 537: consider discussing what would be the range of diversity to be explored, compared to possible screening throughput e.g. via FACS.

Line 554/555: consider elaborating a bit more on how these datasets can be used for modular synthetic genome design

Line 691: please elaborate a bit more on how morphology was selected

Figure 1D: consider to also include the “native” promoter strength for comparison (as mentioned in line 184)

Figure 2C: consider highlighting the observed effects quantitatively especially comparing SCRaMBLE induced SC-HIS (pSCW11) vs SCRaMBLE induced SC-HIS (pRET2)

Figure 2S: consider highlighting band in SC -His

Figure 2C caption: D) consider adding some more details to make self-explanatory, e.g. highlight with and without addition of Histidine

Figure 3G: could be interesting to discuss differences observed comparing yXL327pool and yXL397pool.

Figure S1: good to explain/mention also donor parts

Figure S2: for easing understanding good to consider adding expected gene sizes

Figure S3: please elaborate a bit more on the junctions, expected sizes in the figure and figure captions to facilitate ease of understanding.

Figure S5: C) please mention on what data the percentage was calculated

Figure S7: good to give some background on the different pools and where they are derived from. Please add some more information in the figure and figure captions.

Figure S8: please provide some more details, at what point was sorting done, which sample is shown? Please also explain reasoning behind sub-gating strategies.

Figure S9: please add some more details for ease of understanding e.g. origin of parental genotype for B and C.

General things for discussion:

- 1) What is the theoretical potentially expected diversities generated by SCRaMBLE for the here applied approaches, need to and how to best cover these?
- 2) How to best explain the reduction in duplication phenotype for yXL327 and yXL397. The start is two copies of His5 and then a reduction of Duplication is observed (Figure S10B).
- 3) Highlight novelty in connection to SCRaMble maximizing GFP expression. The authors mention a previous study using SCRaMble where GFP was used as a benchmark.
- 4) Highlight advantages of using the SCOUT system, is there a way to quantify this in terms of how much more efficient the overall method becomes, e.g. need for screening number of variants to cover a certain diversity with and w/o the SCOUT selection.

Reviewer #2

(Remarks to the Author)

Key results

In this work by Lu et al., a new reporter system called SCOUT (SCRaMble Continuous Output and Universal Tracker) was developed in *S. cerevisiae*. This work builds on the Sc2.0 project and relies on the use of SCRaMbled cells containing

either a synthetic chromosome or a synthetic genome module. By combining SCOUT with their previously published long-read sequencing approach (Polar-Seq), the authors achieved high-throughput analysis of the structure and content of induced genome rearrangements, as well as the effects of those arrangements on cell fitness, after iterative SCRaMbLE.

Data, methodology, and validity

The data presented here support the conclusions and claims of the authors; that is, the reporter system seems tractable and useful specifically in the context of the Sc2.0 project. The authors build upon previous work related to the project and present convincing evidence that a combination of high-throughput approaches (use of FACS to sort GFP⁺ cells and long-read sequencing) can provide insight into the outcomes of SCRaMbLE experiments.

Concerns related to data interpretation and conclusions:

While the biological relevance of refactored histidine modules is unclear (see below) additional experiments could improve this aspect of the paper. The authors hypothesize that accumulation of the metabolite IAP is responsible for slow growth phenotypes in SC-His media. If the authors had concrete data to support this hypothesis, it would strengthen their claim that promoter strength affects metabolic flux and is therefore an important consideration in module design. Otherwise, the data in Figure 1 exist solely to provide a testbed (via HISrefactor-4) for SCRaMbLE and SCOUT.

Significance

Compared to previous literature, the advancements described here are 1) development of the SCRaMbLE reporter, SCOUT and 2) coupling of SCOUT to long-read sequencing and FACS to improve selection and characterization of SCRaMbLEd cells. This manuscript contributes new reporter tools specific to the Sc2.0 project but is otherwise limited in scope and biological relevance. SCRaMbLE is an artificial system that can improve growth under various conditions and lead to desired phenotypic outcomes, but in this manuscript, connections made between phenotype and genotype are weak. The utility of iterative SCRaMbLE is also unclear, considering that phenotypes were not significantly improved with additional rounds of SCRaMbLE.

Concerns related to significance:

As demonstrated in Figure 3, SCRaMbLEd HISrefactor-4 cells tend to duplicate HIS5, which significantly improves their growth. Gene duplication is a known adaptive strategy in wild organisms (including humans) and during laboratory evolution experiments. Yet, discussion of these topics is limited in this report. Such a discussion could potentially increase the relevance of refactored HIS modules.

In addition, the refactored HIS module does not seem like a suitable testbed for iterative SCRaMbLE and SCOUT. The authors acknowledge this in the section titled "iterative SCRaMbLE in a synthetic yeast chromosome to maximise phenotype." However, the second test, SCRaMbLE of synthetic chromosome V, does not select for a phenotype that is relevant in biological systems. Instead, the authors choose to screen for cells that maximize GFP expression. These experiments do not provide compelling evidence that iterative SCRaMbLE could be useful in other contexts.

After the chromosome V SCRaMbLE experiment, the authors characterize the SCRaMbLE events and conclude that with the exception of a single gene (PDA1), there is no obvious link to cellular improvements of sfGFP production. The link between PDA1 and increased GFP expression could be explored further to provide convincing evidence of a genotype-phenotype relationship.

While the chromosome V experiment demonstrates that SCRaMbLE and SCOUT may successfully select for the desired phenotype, the weak and potentially non-existent links to genotype are a major concern (see above). It's unclear whether additional experiments of this nature would provide any insights into biological systems.

One interesting chromosome V SCRaMbLE result is the 114 kb inversion and repositioning of the centromere. However, this result is not discussed further in relationship to other recent experiments that reposition the yeast centromere. How did this specific rearrangement affect cell fitness?

In the discussion, the authors mention that SCRaMbLE is an effective solution for creating functional improvements to synthetic modules, and while gene duplication may be a simple solution to the slow growth of HISrefactor-4 it is of limited interest as presented. Changes in the orientation of modular DNA regions in the genome would have provided a more compelling reason to use the system as it could be difficult to achieve such complex rearrangements without SCRaMbLE. Similarly, centromere relocation and other gross chromosomal rearrangements pertaining to chromosome V SCRaMbLE are more interesting from a biological standpoint, but are not discussed.

Version 1:

Reviewer comments:

Reviewer #1

(Remarks to the Author)

All points sufficiently addressed. No further comments.

Reviewer #2

(Remarks to the Author)

The manuscript has been revised successfully. The authors responded to all comments in a meticulous and thoughtful

manner. In particular, the revisions to the discussion created a more compelling and broadly applicable narrative. The authors' rationale for the scope of the work seems justified.

Response to Reviewer Comments

REVIEWER COMMENTS

Reviewer #1 (Remarks to the Author):

The manuscript "Iterative SCRaMbLE for Engineering Synthetic Genome Modules and Chromosomes" by Lu et al. describes iterative SCRaMbIE approaches on a genome module and chromosome level. A novel concept for targeted selection and analysis of scrambled genome modules named SCOUT was introduced. The method is well substantiated with respective data sets and was applied on a histidine biosynthesis genome module as a proof of concept. The methodology has novelty and will provide the community with a valuable new tool to apply SCRaMbIE based methods in more straight forward iterative manner, paving the way for pathway and genome diversification and screening. Some additional considerations and motivation for conducted experiments would benefit the coherence of the study. For example, it would have been nice to see the application of the SCOUT system on the chromosome level. Did the authors consider this and if not why? Also, novelty aspects of the study should be highlighted and benchmarked a bit more compared existing studies.

We thank the reviewer for their time and expertise in assessing our manuscript. We were very pleased to read their kind praise of SCOUT as a novel and valuable tool for iterative SCRaMbLE applications. We agree that applying SCOUT at the chromosomal scale is a promising future direction. In this study, we focused on the genome module level to show a proof of concept that is easily validated both by genotype and phenotype analysis. SCRaMbLing the genome modules (genomically integrated regions that are <35 kb) allowed us to identify SCRaMbLE diversity as a library using our PCR-based POLAR-seq method. Extending to the chromosomal level makes targeted sequencing a lot more difficult due to the amplicon size limitation of PCR-based library sequencing and lack of tailored strategies for selective enrichment of specific chromosomes. We've been exploring chromosome-specific enrichment sequencing methods and developing SCOUT-based tools for larger-scale applications, but these methods are very different to those in this paper, so hopefully will be reported in a separate study soon. We have revised the discussion section to highlight the unique advantages of SCOUT compared to traditional colony screening, including screening throughout, diversity coverage, time and cost effectiveness. It will be valuable for complex genome engineering projects where iterative SCRaMbLE can be applied.

Further considerations and remarks are summarized as follows:

Line 101: could be a more explicit what "rearranging genes within this module" would entail (e.g. also including duplications).

This sentence has been reworded. See line 102.

Line 123-127: For clarity one could mention here already that loxP sites are also included.

We have revised the sentence to clarify the inclusion of loxP sites. See lines 122-123.

Line 142: General comment, good to introduce abbreviation when used first time, e.g. here YTK. Could also consider having a list of relevant abbreviations (e.g. for MYT, IAP, etc.)

We thank the reviewer for pointing this out. We've now introduced the abbreviation "YTK" at its first mention and have added a list of relevant abbreviations, including MYT and IAP, etc. See line 144, 219-220, 267.

Line 191/192: good to make a bit clearer the motivation behind assigning the strongest promoter for HIS3 and the weakest promoter assigned for HIS5

Thanks for the suggestion. We have added more clarification about the motivation behind promoter selection for the *HIS* genes. See line 195-196. *HIS3* was assigned the strongest promoter because it showed the highest transcript level based on RNA-seq data under YPD conditions (Reference 28). While *HIS5* with the lowest transcript level, was assigned the weakest promoter.

Line 205-210: Would be good to elaborate a bit more why HISrefactor-4 behaves so much differently from low strength-3 when it comes to growth. HIS3 and HIS5 difference in expression level seems to be similar.

Thanks for the suggestion. We have added more elaboration about the behaviour difference between mixed *HIS*_{refactor-4} and low-strength *HIS*_{refactor-3} strains. See lines 221-224. The difference in promoter strength between *HIS3* and *HIS5* in *HIS*_{refactor-3} is relatively smaller due to the use of the *pRPL18b* promoter for *HIS3* and the *pREV1* promoter for *HIS5*, which have only ~30-fold relative difference, as reported in the original Lee et al. YTK paper.

Line 233: "excises one loxP-variant" instead of "excises of one loxP-variant"

We thank the reviewer for pointing this out. We have edited in line 240 as the reviewer suggested.

Line 241 onwards: good to introduce the estradiol inducibility of the system to ease overall understanding

Thanks for the suggestion. We've now added the estradiol inducibility in line 237.

Line 276-278: consider rephrasing "constructed for Figure 1", and be a bit more explicit, e.g. which

cell with which type of module was chosen?

Thanks for the suggestion. We've now rephrased this sentence. See lines 287-288.

Line 303: Were any point mutations observed in the sequence data?

We thank the reviewer for this point. We didn't specifically analyse point mutations, as our aim was to quickly identify structural variations generated by SCRaMbLE such as genes and gene combinations rather than single-nucleotide variation. Our analysis pipeline tolerates up to 12.6% base-level errors, allowing for potential point mutations without compromising gene rearrangement detection (Reference 26). Although base mutations may be present, they are unlikely to influence our overall conclusions at the resolution we target.

Line 315: Could the deletion be connected to that the entire module was relocated? A simple PCR targeting parts of the module could give insights if it is still present.

Sorry about the confusion on this point. The deletions we observed here are deletions of some specific genes within the module rather than whole module deletion, as shown by intact reads obtained from long range PCR and long read sequencing. These reads identified from sequencing are full barcoded amplified DNA of the entire module that is still present at the original *URA3* locus. It would be difficult to isolate these deletion genotypes due to their low abundance (<1%) within the library. We've now added clarification to lines 324-325 to better explain the deletion.

Line 319: Is enrichment of shorter amplicons what you observed here?

We did not specifically observe enrichment of shorter amplicons in our data. Here, we recommend considering the possibility of PCR length amplification bias when estimating genotype abundance using the POLAR-seq method, which is based on long-range PCR and long read sequencing.

Line 321: The approach of picking random colonies would not give direct insights into all relevant genotypes observed using POLAR-seq. Why not choosing a strategy to analyze the most represented genotypes and evaluate these instead of randomly picking some (e.g. picking representative examples of each type shown in Fig.3 D)?

In addition, cleaner experiments would have been to reintroduce the modified His module into the initial strain. This would exclude potential secondary effects e.g. in connection to random mutations in the genome. Was this considered?

We thank the review for the suggestions. The aim of random colony picking is to confirm whether genotype abundance identified in POLAR-seq is consistent with the single cell screening. This helps us confirm the read frequency in our POLAR-seq analysis is representative of the number of cells

with that genotype in the sampled pool. We've now clarified this in line 351-352. It would be difficult to pick out every type shown in Fig 3D, as some of those sequenced at <1-2% will be at very low frequency in the library and would require picking and individually sequencing thousands of colonies to identified.

As suggested by the reviewer, we have now included additional experimental data showing the effect of introducing a second copy of the *HIS5* gene driven by the same weak *RAD27* promoter at a different genomic locus in the strain with poor growth on media lacking histidine. This second copy restored the selective growth fitness in the absence of histidine, demonstrating that the fitness defect was due to the lack of *HIS5* expression. Supporting data are now included in Fig S10 and revised text in line 363-368.

Line 329 and Fig. S9A: Only 8/14 of the clones were further investigated in terms genotype. Further insights on 1-4, 1-5, 1-6, 1-7, 1-9 and 1-14 would be interesting to discuss as well. Especially for the ones which were scoring with relatively high growth rates in Fig. 3E SC-His (e.g. 1-14).

We tried to isolate gDNA, perform long-range PCRs and sequence all 14 clones. Unfortunately, 6 out of 14 (including clone 1-14) failed in one of these steps which prevented us from obtaining complete genotype data for all the clones. While disappointing, we believe that the consistent genotypes we observe across the 8 successfully analysed clones is sufficient evidence to show that the relative abundance of genotypes we observe in our POLAR-seq data are representative of our library.

Line 420: Generally, why was not FACS considered to enrich for high FL variants? By randomly picking colonies, not the full scope of variants can be screened/analyzed. Could higher FL variants have been identified?

We thank the reviewer for the suggestion. To clarify, in line 420 (now line 439), the 7 colonies we randomly picked were from plates used to remove the pSCW11-creEBD plasmid. In this experiment, random colony picking was to avoid potential bias toward variants with ongoing Cre activity, which may lead to continuing SCRaMbLE if the plasmid had not been fully lost. These 7 isolates represent biological replicates of single strains to measure increase in GFP production over rounds.

We agree that FACS-based enrichment could help enrich high-FL variants by screening a larger and more diverse population. We have investigated this approach in a separate study, which will be published soon. For synthetic chromosome SCRaMbLE, targeted enrichment and cost-effective deep sequencing of specific chromosomes from pooled, FACS-sorted cell populations remains a big technical challenge. Therefore, we prioritised whole genome sequencing of a small number of well-performing isolates rather than larger scale library screening to reveal genotype-phenotype relationship across iterative rounds of synthetic chromosome V SCRaMbLE. Combining FACS enrichment with pooled targeted chromosome sequencing is a future direction we are investigating.

Line 487: Consider rephrasing “synthetic promoters” as the promoters used here are not really “synthetic”

We've now reworded this sentence by replacing the “synthetic promoters” with YTK promoters. See line 509.

Line 496: explain abbreviation AICAR

We've now added explanation of AICAR. See line 518-519.

Line 509: introduce abbreviation GEMbL2R

We've now added introduction of GEMbL2R. See line 532.

Line 537: consider discussing what would be the range of diversity to be explored, compared to possible screening throughput e.g. via FACS.

We thank the reviewer's suggestion on discussing the relationship between library diversity and screening throughput via FACS. In our *HIS* module design, recombination between 8 loxP sites can theoretically generate ~1 million distinct genotypes considering only deletions and inversions. While duplication genotypes could further increase this diversity, it is difficult to estimate the upper limit of duplications. In our study, the actual diversity is expected to be much lower due to the enrichment for faster-growing genotypes under histidine absent selection, in which deletion genotypes were diluted out. We collected >1 million possible SCRaMbLEd GFP⁺ cells via FACS, which should cover a large portion of the diversity space. Collecting more cells via FACS and reducing following growth time may help cover more diversity and minimise potential bias, depending on experimental purposes. In response to this feedback, we've now added discussion of library size and screening throughput into the revised manuscript. See lines 586-594.

Line 554/555: consider elaborating a bit more on how these datasets can be used for modular synthetic genome design

We thank the reviewer for the suggestion. We've now added a few sentences to clarify more on how datasets obtained from SCRaMbLE can be used for modular synthetic genome design. See lines 658-667.

Line 691: please elaborate a bit more on how morphology was selected

We have added more details about how yeast morphology was selected. Specifically, in the initial

gating step (FSC-A vs SSC-A), we selected for intact, healthy yeast cells with characteristic yeast cell size while excluding debris. The main population as a dense cluster was gated first. See lines 810-812.

Figure 1D: consider to also include the “native” promoter strength for comparison (as mentioned in line 184)

We've now revised Figure 1D to include transcript level of native *HIS* genes based on the RNA-seq data under YPD conditions (Reference 28) for comparison.

Figure 2C: consider highlighting the observed effects quantitatively especially comparing SCRaMBLE induced SC-HIS (pSCW11) vs SCRaMBLE induced SC-HIS (pRET2)

Thanks for the suggestion. We've now included quantitative data showing the percentage of GFP+ cells after 4 hours of induction and number of GFP+ colonies after 3 days of plating in the new figure 2 for comparison, including SCRaMBLE induced SC-HIS (pSCW11) and SCRaMBLE-induced SC-HIS (pRET2).

Figure 2S: consider highlighting band in SC -His

We've now highlighted the band in SC-His in the revised Figure 2D.

Figure 2C caption: D) consider adding some more details to make self-explanatory, e.g. highlight with and without addition of Histidine

We've now added more details to caption (D), highlighting with and without histidine. See lines 263-264.

Figure 3G: could be interesting to discuss differences observed comparing yXL327pool and yXL397pool.

We've now added a short discussion comparing yXL327pool and yXL397pool. See lines 401-404.

Figure S1: good to explain/mention also donor parts

We've now added explanation of donor parts in Fig S1 caption. See lines 1021-1022.

Figure S2: for easing understanding good to consider adding expected gene sizes

We've now added expected amplicon lengths for junction PCRs in the Fig S2A and C. See lines 1030-

1031.

Figure S3: please elaborate a bit more on the junctions, expected sizes in the figure and figure captions to facilitate ease of understanding.

We've now added expected amplicon sizes for junction PCRs to confirm refactored module integration in Fig S3 and its caption. See lines 1037-1039.

Figure S5: C) please mention on what data the percentage was calculated

We've now clarified in Fig S5 caption that the percentage was calculated based on flow cytometry data shown in panel B, using a gating strategy defined by a negative control. See lines 1058-1059.

Figure S7: good to give some background on the different pools and where they are derived from. Please add some more information in the figure and figure captions.

We've now added background information about the different pools and where they are from in both the supplementary figure and figure legend. See lines 1081-1082.

Figure S8: please provide some more details, at what point was sorting done, which sample is shown? Please also explain reasoning behind sub-gating strategies.

We've now provided more details as the reviewer suggested in the figure caption. See lines 1085-1089.

Figure S9: please add some more details for ease of understanding e.g. origin of parental genotype for B and C.

We've now added more details about origin of parental genotype for Fig S9B and C in both the figure and figure caption. See lines 1096-1097.

General things for discussion:

1) *What is the theoretical potentially expected diversities generated by SCRaMBLE for the here applied approaches, need to and how to best cover these?*

We thank the reviewer's suggestion on discussing the relationship between library diversity and screening throughput via FACS. In our *HIS* module design, recombination between 8 loxPsym sites can theoretically generate ~1 million distinct genotypes considering only deletions and inversions. While duplication genotypes could further increase this diversity, it is difficult to estimate the upper

limit of duplications. In our study, the actual diversity is expected to be much lower due to the enrichment for faster-growing genotypes under histidine absent selection, in which deletion genotypes were diluted out. We collected >1 million possible SCRaMbLEd GFP⁺ cells via FACS, which should cover a large portion of the diversity space. Collecting more cells via FACS and reducing the following growth time may help cover more diversity and minimise potential bias, depending on experimental purposes. We've now added a discussion of library size and screening throughput in the revised manuscript. See lines 585-594.

2) *How to best explain the reduction in duplication phenotype for yXL327 and yXL397. The start is two copies of His5 and then a reduction of Duplication is observed (Figure S10B).*

The *HIS5* duplication frequency in the reads identified from yXL327pool and yXL397pool is 85.53% and 93.48%, respectively. Compared to the 86.89% observed in the yXL219pool in the first round of SCRaMbLE, the slight decrease in *HIS5* duplication frequency in yXL327pool is mostly because of the increase in reads identified with *HIS5* triplications and quadruplications (Figure 3F and G).

3) *Highlight novelty in connection to SCRaMbLE maximizing GFP expression. The authors mention a previous study using SCRaMbLE where GFP was used as a benchmark.*

We thank the reviewers for this point. In the previous study we referenced, the GFP expression phenotype was used to investigate whether the increased expression of a metabolic pathway observed after a *single* round of SCRaMbLE was caused by specific mutations or rearrangements generated by SCRaMbLE. However, here we use GFP expression as a benchmark to assess how chromosome-wide modifications across multiple iterative rounds of SCRaMbLE can enhance the heterologous expression of a single protein. Unlike the case of *HIS* module, this is a simple phenotype that is not directly linked to growth and doesn't have an obvious solution. This proxy allows us to evaluate applying iterative SCRaMbLE for optimising yeast as a cell factory for heterologous production of other proteins of interest, such as metabolic production and protein secretion, etc. We have added this information into our revised discussion section. See lines 596-605.

4) *Highlight advantages of using the SCOUT system, is there a way to quantify this in terms of how much more efficient the overall method becomes, e.g. need for screening number of variants to cover a certain diversity with and w/o the SCOUT selection.*

We thank the reviewer for the suggestion. A direct benchmarking experiment comparing the SCOUT system with conventional colony-based screening would be helpful for quantifying the efficiency. However, a full comparative analysis of diversity coverage with and without SCOUT selection would require large-scale deep sequencing across multiple individual isolates/pools, which is beyond the scope of this study. Here, we focus on establishing and validating the SCOUT-based workflow as a reliable tool for genotyping SCRaMbLEd yeast libraries.

Our current data suggest that the SCOUT system improves screening efficiency. Additional flow cytometry data in Fig 2C revealed that after 4 hours of induction, ~50% of the cells exhibited GFP fluorescence indicating possible SCRaMbLE events. This suggests that the SCOUT may increase the screening efficiency by at least 2-fold within this induction period when no obvious phenotype changes are observed. We also demonstrate the screening efficiency of SCOUT by showing 4 random picked GFP+ colonies without FACS-sorting all showed rearrangements (Fig S6) and the consistent recovery of dominant genotypes (~37.5%) directly from FACS-sorted pools Fig S9).

A big advantage of the SCOUT system is that it significantly reduces the time and cost of identifying genotypes from large and diverse SCRaMbLEd populations. Unlike conventional single colony screening methods, SCOUT streamlines the workflow by avoiding colony plating, picking, and culturing. This saves 3 to 4 days. When combined with long-read sequencing, SCOUT enables cost-effective, high-throughput genotype analysis by directly identifying yeast cluster genotypes from FACS-sorted libraries at a cost as low as \$1.5 per genotype (Reference 26). In addition, SCOUT facilitates the collection and enrichment of millions of SCRaMbLEd cells in a single FACS run, providing broader coverage of library diversity compared to previous colony-based screening methods. This higher sampling depth increases the chance of capturing rare rearrangement genotypes such as those with read frequencies below 1%, which might be missed due to the throughput limit of traditional screening. We have included this information in the revised discussion section. See lines 572-594.

Reviewer #2 (Remarks to the Author):

Key results

*In this work by Lu et al., a new reporter system called SCOUT (SCRaMbLE Continuous Output and Universal Tracker) was developed in *S. cerevisiae*. This work builds on the Sc2.0 project and relies on the use of SCRaMbLEd cells containing either a synthetic chromosome or a synthetic genome module. By combining SCOUT with their previously published long-read sequencing approach (Polar-Seq), the authors achieved high-throughput analysis of the structure and content of induced genome rearrangements, as well as the effects of those arrangements on cell fitness, after iterative SCRaMbLE.*

Data, methodology, and validity

The data presented here support the conclusions and claims of the authors; that is, the reporter system seems tractable and useful specifically in the context of the Sc2.0 project. The authors build upon previous work related to the project and present convincing evidence that a combination of high-throughput approaches (use of FACS to sort GFP+ cells and long-read sequencing) can provide insight into the outcomes of SCRaMbLE experiments.

We thank the reviewer for their enthusiasm on synthetic yeast genome project and expert insights into the SCRaMbLE system that has helped us to revise the manuscript. We are very pleased that the reviewer recognised SCOUT as a useful reporter system within the Sc2.0 framework and found our combination of high-throughput approaches convincing in elucidating the structural and functional consequences of iterative SCRaMbLE. Our point-to-point reply to the concerns are as below.

Concerns related to data interpretation and conclusions:

While the biological relevance of refactored histidine modules is unclear (see below) additional experiments could improve this aspect of the paper. The authors hypothesize that accumulation of the metabolite IAP is responsible for slow growth phenotypes in SC-His media. If the authors had concrete data to support this hypothesis, it would strengthen their claim that promoter strength affects metabolic flux and is therefore an important consideration in module design. Otherwise, the data in Figure 1 exist solely to provide a testbed (via HISrefactor-4) for SCRaMbLE and SCOUT.

We thank the reviewer for the comment on the biological relevance of the refactored histidine pathway and the role of promoter strength in modulating metabolic flux. To address this, we have performed additional experiments to strengthen our hypothesis that the observed growth defect in SC-His medium results from an imbalance in metabolic flux due to significant difference in enzyme expression between *HIS3* (driven by a very strong promoter) and *HIS5* (by a very weak promoter). In our new work we introduced a second copy of *HIS5* driven by the same weak *RAD27* promoter at a different genomic locus into the original *HIS*_{refactor-4} strain that showed poor growth. Results showed that this restored growth fitness in the absence of histidine, directly supporting the simple explanation that

adjusting expression strength of *HIS5* alleviates a metabolic bottleneck and directly influences metabolic flux. This additional new data is now included as Fig S10. This is described in revised text on lines 363-368.

Significance

Compared to previous literature, the advancements described here are 1) development of the SCRaMbLE reporter, SCOUT and 2) coupling of SCOUT to long-read sequencing and FACS to improve selection and characterization of SCRaMbLEd cells. This manuscript contributes new reporter tools specific to the Sc2.0 project but is otherwise limited in scope and biological relevance. SCRaMbLE is an artificial system that can improve growth under various conditions and lead to desired phenotypic outcomes, but in this manuscript, connections made between phenotype and genotype are weak. The utility of iterative SCRaMbLE is also unclear, considering that phenotypes were not significantly improved with additional rounds of SCRaMbLE.

We thank the reviewer for the comments on the advancements of our SCOUT-POLAR-seq approach to enable high-throughput sequencing and quantitative analysis of post-SCRaMbLE cell libraries. We also thank the reviewer's interest in the genotype–phenotype relationship and the potential utility of iterative SCRaMbLE.

While we agree that mechanistic links of SCRaMbLE events to phenotype would be of great interest, we have learned over the decade that we have worked with SCRaMbLE and synthetic yeast, that this turns out to be a major task for every isolated strain – usually requiring an expensive multi-omics approach for each genotype-to-phenotype observation of interest. We believe that this kind of challenging work is beyond the scope of this manuscript and would additionally distract from our main focus in this paper which is using high-throughput screening to identify diverse genotypes from SCRaMbLE that inform module and genome design strategies. We provide speculation on how specific gene rearrangements might contribute to phenotype improvements. While this is not our focus, we hope including this speculation can provide ideas for other future studies on how SCRaMbLE can generate unpredictable targets for optimising a phenotype with non-obvious genetic solutions. Sadly, cost-effective methods that can determine genotype to phenotype mechanistic connections at scale from SCRaMbLE remain elusive and are a long term goal for our group and others.

However, in light of the reviewer comments we have now included additional data to strengthen the connections between phenotype and genotype, with Figs S9 and S10 now supporting that SCRaMbLE-informed genotype solutions can effectively balance metabolic flux by tuning gene expression. These data highlight that our approach can not only be used for phenotype optimisation, but also for identifying specific genotypic solutions for improving desired functional outcomes.

Regarding the utility of iterative SCRaMbLE, while our work showed that phenotypic improvements rapidly plateaued after the second SCRaMbLE round on a genome module, we observed convergent

genotypic solutions, such as *HIS5* duplication continued in the second round, generating triplication and quadruplication. This suggests that the system consistently explores a defined fitness landscape, producing reproducible and functionally relevant outcomes even when the phenotypic gain is modest. We observed that the first round of SCRaMbLE yielded strains with significant phenotypic improvements, but subsequent rounds displayed a rapid plateau. This is consistent with iterative-SCRaMbLE committing to a single local maximum in the phenotype-fitness landscape. We emphasised that this maximum may not be the global optimum and that it could be limited by the scope of the rearrangements done in the first round. We provide recommendations on maximising diversity in the first round by increasing the screening population size. Our findings also indicate that the optimal SCRaMbLE iterations may be context-dependent, influenced by genomic contexts and the specific traits being optimised. This highlights that iterative SCRaMbLE is not a one-size-fits-all solution, but rather a tailored strategy depending on the biological context. By clarifying these points, we hope to highlight that while iterative SCRaMbLE may not generate significant phenotypic improvements in every round, it is still valuable for identifying diverse genetic solutions and exploring the evolutionary dynamics within a defined fitness landscape.

Concerns related to significance:

As demonstrated in Figure 3, SCRaMbLEd HISrefactor-4 cells tend to duplicate HIS5, which significantly improves their growth. Gene duplication is a known adaptive strategy in wild organisms (including humans) and during laboratory evolution experiments. Yet, discussion of these topics is limited in this report. Such a discussion could potentially increase the relevance of refactored HIS modules.

We thank the reviewer for the suggestion. We've added the discussion of gene duplication as an adaptive strategy in our revised manuscript. See lines 631-648.

In addition, the refactored HIS module does not seem like a suitable testbed for iterative SCRaMbLE and SCOUT. The authors acknowledge this in the section titled "Iterative SCRaMbLE in a synthetic yeast chromosome to maximise phenotype." However, the second test, SCRaMbLE of synthetic chromosome V, does not select for a phenotype that is relevant in biological systems. Instead, the authors choose to screen for cells that maximize GFP expression. These experiments do not provide compelling evidence that iterative SCRaMbLE could be useful in other contexts.

We thank the reviewer for this point. Our study of the refactored *HIS* module and SCRaMbLE of synthetic chromosome V provide insights into the dynamics of iterative SCRaMbLE across different genomic contexts, including both module- and chromosome-scale rearrangements as well as under selective and non-selective conditions. Compared with previous studies that focused on optimising a specific trait through certain rounds of SCRaMbLE, we explored a broader conceptual and technical aspects by showing improvement space each round of SCRaMbLE has and the potential of SCRaMbLE not only as an optimisation tool but also as a model for adaptive genome evolution. See

our revised discussion at lines 562-570.

The *HIS* module is actually a good testbed as it allows us to examine how fast growth-related phenotype improvements plateau and how the genotypic diversity generated by rearranging 7 genes from a *conditionally* essential pathway varies with or without selection. These are new aspects that haven't been explored in previous studies. Our SCOUT-POLAR-seq pipeline enables quantitative mapping of genotype abundance, providing a way of estimating the improvement space of each round of SCRaMbLE and identifying iterative cycle at which these improvements plateau. This is now further emphasised with discussion in lines 562-570.

Regarding SCRaMbLE of synthetic chromosome V and our focus on GFP expression, we intentionally chose this phenotype as a simple model to explore iterative SCRaMbLE in optimising gene expression for heterologous protein production. Although GFP expression does not directly represent a specific biologically relevant phenotype, it is used as a benchmark to track how chromosomal rearrangements can enhance heterologous protein expression, which is the major goal in many industrial yeast biotechnology projects. This approach serves as a proxy for optimising other functional traits, such as metabolic production and protein secretion. See additional discussion in lines 596-605.

After the chromosome V SCRaMbLE experiment, the authors characterize the SCRaMbLE events and conclude that with the exception of a single gene (PDA1), there is no obvious link to cellular improvements of sfGFP production. The link between PDA1 and increased GFP expression could be explored further to provide convincing evidence of a genotype-phenotype relationship.

We thank the reviewer for this point. We apologise that our current statement was not very clear. We've added more explanation in lines 491-495 and in the discussion section. While we agree that elucidating the mechanistic links of genotype to phenotype would be valuable, we feel this work is beyond the scope of this manuscript whose main focus is showcasing how high-throughput screening can identify diverse genetic solutions from SCRaMbLE that inform module and genome design strategies. Our observation of *PDA1* gene duplication from synV SCRaMbLE was also observed in past work, in a synV SCRaMbLEd strain (BA03) that showed improved betulinic acid synthesis (Reference 24). However, it is not immediately clear to us whether *PDA1* duplication is directly linked to metabolic flux changes that improve heterologous protein production or whether the flanking loxPsym sites are just more likely to be involved in SCRaMbLE events. We include this speculation to provide some ideas for future studies and have added clarification in the discussion at lines 608-614.

While the chromosome V experiment demonstrates that SCRaMbLE and SCOUT may successfully select for the desired phenotype, the weak and potentially non-existent links to genotype are a major concern (see above). It's unclear whether additional experiments of this nature would provide any insights into biological systems.

We acknowledge that the mechanistic link is not immediately clear, especially for the synV SCRaMbLE experiments. While further investigation to establish this link would be valuable, our focus is to present a proof-of-concept framework that demonstrates using SCRaMbLE and SCOUT to perform large scale screening to track the diversity and abundance of genotypes across iterative SCRaMbLE rounds. Delving into phenotypes and how novel genotypes link to these is a major undertaking that we do not focus on in this paper. Instead, we hope that the methods we describe here are useful for generating hypotheses and that identifying diverse genetic solutions can guide more targeted mechanistic studies in the future. Future studies combining transcriptomic, proteomic and metabolomics analyses may help systematically map the genetic changes associated with observed phenotypic traits, thereby providing deeper insights into how specific SCRaMbLE rearrangements contribute to functional outcomes. We've now added these points into the discussion section as mentioned before.

One interesting chromosome V SCRaMbLE result is the 114 kb inversion and repositioning of the centromere. However, this result is not discussed further in relationship to other recent experiments that reposition the yeast centromere. How did this specific rearrangement affect cell fitness?

We thank the reviewer for the suggestion. We've now added the discussion of centromere relocation event in lines 614-623.

In the discussion, the authors mention that SCRaMbLE is an effective solution for creating functional improvements to synthetic modules, and while gene duplication may be a simple solution to the slow growth of HISrefactor-4 it is of limited interest as presented. Changes in the orientation of modular DNA regions in the genome would have provided a more compelling reason to use the system as it could be difficult to achieve such complex rearrangements without SCRaMbLE. Similarly, centromere relocation and other gross chromosomal rearrangements pertaining to chromosome V SCRaMbLE are more interesting from a biological standpoint, but are not discussed.

We thank the reviewer for the suggestion and agree with the points being made. We've now revised the manuscript to discuss effects of inversions of modular DNA regions and centromere repositioning, see lines 614-623 and 643-648 for this text.